**METHOD**

# Towards in silico CLIP-seq: predicting protein-RNA interaction via sequence-to-signal learning

Marc Horlacher[1,2,3,4*] , Nils Wagner[3,4], Lambert Moyon[1], Klara Kuret[5,6,7], Nicolas Goedert[1], Marco Salvatore[2], Jernej Ule[5,6], Julien Gagneur[1,3,4], Ole Winther[2*] and Annalisa Marsico[1,4*]

*Correspondence:
marc.horlacher@helmholtz-muenchen.de; ole.winther@bio.ku.dk; annalisa.marsico@helmholtz-muenchen.de

[1] Computational Health Center, Helmholtz Center Munich, Munich, Germany
[2] Department of Biology, University of Copenhagen, Copenhagen, Denmark
[3] Department of Informatics, Technical University of Munich, Garching, Germany
[4] Helmholtz Association - Munich School for Data Science (MUDS), Munich, Germany
[5] National Institute of Chemistry, Ljubljana, Slovenia
[6] The Francis Crick Institute, London, UK
[7] Jozef Stefan International Postgraduate School, Jamova cesta 39, 1000 Ljubljana, Slovenia

## Abstract

We present RBPNet, a novel deep learning method, which predicts CLIP-seq cross-link count distribution from RNA sequence at single-nucleotide resolution. By training on up to a million regions, RBPNet achieves high generalization on eCLIP, iCLIP and miCLIP assays, outperforming state-of-the-art classifiers. RBPNet performs bias correction by modeling the raw signal as a mixture of the protein-specific and background signal. Through model interrogation via Integrated Gradients, RBPNet identifies predictive sub-sequences that correspond to known and novel binding motifs and enables variant-impact scoring via in silico mutagenesis. Together, RBPNet improves imputation of protein-RNA interactions, as well as mechanistic interpretation of predictions.

**Keywords:** CLIP-seq, Deep learning, Computational biology, Protein-RNA interaction

## Background

RNA-binding proteins (RBPs) are a family of proteins that bind to coding and non-coding transcripts, usually through recognition of short sequence or structural features commonly known as motifs [10]. To date, over 2, 000 proteins have been experimentally identified as RNA-binding, rendering it one of the largest cellular protein groups [17]. RBPs are involved in every aspect of post-transcriptional regulation, including modification, stabilization, localization, splicing, and translation of RNAs [25]. Misregulation of RBPs, as well as mutations in their amino acid sequence or the sequence of their RNA targets, has been associated with an abundance of human diseases, including neurological and psychiatric disorders [49]. Therefore, uncovering binding preferences and RNA targets of RBPs is crucial for understanding the role of RBPs in post-transcriptional regulatory pathways and for quantifying the impact of their dis-regulation in context of human disease. Nowadays, the most common protein-centric experimental approach to profile RNA-binding in vivo is via individual-nucleotide resolution Cross-Linking and

Immunoprecipitation followed by sequencing (CLIP-seq) [29, 61] and its derivatives, which enables transcriptome-wide detection of protein-RNA interactions for a protein of interest. Variants of CLIP-seq, including *individual-nucleotide* and *enhanced* CLIP (iCLIP and eCLIP, respectively), further allow for detection of protein-RNA crosslinking events at single-nucleotide resolution. Here, in brief, cells are radiated with UV light, forming covalent cross-links between RNA molecules and bound proteins. Protein/RNA complexes are then purified with protein-specific antibodies and the bound proteins are digested. Subsequently, the bound RNA molecules are reverse-transcribed to cDNA, followed by high-throughput sequencing. Reverse-transcription often truncates at the cross-linked site due to a small peptide remaining bound to the site after protein digestion. After alignment of reads to the reference genome, this leads to an accumulation of read-starts at one nucleotide down-stream of the cross-linked site. The resulting nucleotide-wise count signal can then be used to identify binding sites of the RBP of interest as RNA regions where the signal is significantly higher than expected, given some background model [57]. CLIP-seq data is commonly post-processed with peak callers, which identify a number of regions of enriched signal (also referred to as *binding sites*), usually in the order of thousands. As peak calling is subject to unspecific cross-linking events in the underlying data, inferring target-specific signal from CLIP data is crucial. Multiple studies have identified a number of CLIP-associated biases, including background signal from abundant RNAs that are not properly washed during library preparation, library contamination with bound RNAs of other RBPs and strong UV crosslinking bias towards single-stranded uridine-rich motifs [22]. CLIP-seq experiments are often paired with a control, to account for unspecific background signal. For instance, the eCLIP protocol is designed to generate a size-matched input (SMInput) control by omitting the IP step such that the resulting library represents RNA fragments crosslinked to a mixture of background proteins with a similar molecular weight as the target RBP. Therefore, eCLIP SMInput data is a powerful resource to correct computational methods for experimental bias and reduce the number of detected unspecific binding events for an RBP of interest.

Analysis of experimentally identified RNA-binding sites can give insight into both the functional role of the RBP as well as the RNA sequence and structure feature by which it identifies (and binds to) its RNA targets. In addition, knowledge of the binding preference of an RBP enables imputation of protein-RNA interaction on RNAs not present at the time of the experiment. While traditional methods for de novo motif finding [3] or more sophisticated generative models [24] analyze identified RNA-binding sites directly, for instance by aggregating enriched sub-sequences into motif position-weight matrices (PWMs), more recent model-based approaches [32, 44] attempt to model RBP-binding as a function of RNA sequence for a given protein of interest. That is, model-based approaches attempt to find a function of the form $f : RNA \rightarrow \sigma(CLIP)$, where $\sigma$ is some post-processing function on the raw experimental CLIP, for instance a peak caller. Model-based analysis of RBP binding has several key advantages over traditional model-free approaches. First, the availability of a model allows for imputation of missing binding site information. As only a fraction of transcripts may be expressed in the experimental cell type at a given time, CLIP-seq experiments generally draw an incomplete picture of an RBP's binding landscape. Here, predictions from RBP-binding models may aid researchers in generalizing their analysis to unobserved transcripts by providing

candidate sites of protein-RNA interaction. The ability to impute missing binding information on arbitrary RNA sequences is especially relevant for the analysis of RBP-binding to foreign sequences, such as foreign RNAs derived from viruses [27]. Second, the model *f* represents a simplified abstraction of the in vivo biology, that is, the recognition of a binding site in a target RNA sequence by the protein of interest. Besides identifying drivers of RNA target recognition (e.g., binding motifs), the ability to study *f* enables "what-if" analysis, allowing one to explore how changes in the RNA input sequence affect RBP binding. For instance, researchers may investigate the impact of single-nucleotide variants (SNVs) on RBP binding in silico.

Deep learning enabled ground-breaking performance on tasks across a broad domain of research, including the computational modeling of protein-RNA interaction [1, 6, 18, 48]. Current state-of-the-art RBP-binding predictions models are generally classification-based, that is, given an input RNA sequence, the model is tasked with predicting whether the sequence is *bound* or *unbound* by the protein of interest. While classification-based models represent a significant improvement over traditional methods, they have several limitations. First, training and evaluation of classification-based models requires prior annotation of sequences with high-quality binary bound/unbound labels, generally through the use of peaks callers, making the model heavily dependent on upstream preprocessing steps. Therefore, performance of classification-based models is highly sensitive towards the availability of unbiased bona fide binding sites. Second, predictions of classification-based models generally have low resolution. While methods commonly take as input RNA sequences of 100s of nucleotides in length, the predicted label is assigned to the entire input region. This create ambiguity with regards to the exact location of the protein-RNA interaction site, which usually spans only a few nucleotides. Third, binary labels (*bound* and *unbound*) modeled by classification-based methods represent a strong simplification of the information yielded by CLIP-seq experiment. Compression of the CLIP-seq signal footprint within transcript region to a binary value may lead to loss of essential information for understanding the nuances of protein-RNA interaction.

Recently, a new class of models has emerged, which directly predict experimental signal from genomic sequences [2, 34]. For instance, BPNet [2] trains a dilated convolutional neural network which models transcription factor (TF) binding by predicting ChIP-nexus signal from DNA sequences at base resolution. However, there is a lack of sequence-to-signal models for prediction tasks on RNA sequences, including the task of modeling protein-RNA interaction via CLIP-seq signal prediction. In context of CLIP-seq, the presence of technical biases and cell-type specific RNA abundance pose a challenge with respect to the identification of sequence-mediated binding mechanisms at single-nucleotide resolution. Therefore, models need to account for the fact that the observed signal may partially be observed due to technical biases, rather than protein-RNA interaction of target RBP. This effect may further depend on the RNA sequence context, as is the case for nucleotide-specific crosslinking biases or sample contamination with other RBPs which themselves have certain sequence-depended binding preferences.

To fill this gap we developed RBPNet, a sequence-to-signal dilated convolutional neural network, which learns a direct mapping of RNA sequences to crosslink count signal

extracted from CLIP-seq experiments. Given a variable-length RNA input sequence, RBPNet predicts the *distribution* of crosslink counts at single-nucleotide resolution along the input sequence, thereby enabling learning the binding profile of an RBP on a transcripts of arbitrary length. Existing sequence-based models of CLIP data are binary classifiers which necessitate important preprocessing steps, thus loosing the high resolution and the quantitative nature of the data In contrast to current state-of-the-art classifiers, RBPNet does not require a peak caller to produce transcript regions as candidate sites for model training. Instead, it uses a lenient cutoff-based approach to train on all regions that show an enrichment of crosslink counts, thereby making maximal use of signal generated by the underlying experiment. By additionally modeling the background signal of a paired control experiment, RBPNet implicitly learns the bias and protein-specific crosslinking components of the total signal, allowing one to disentangle genuine signal from noise. By training on hundreds of thousands of regions per RBP CLIP experiment, RBPNet reaches high accuracy in predicting RBP-binding signal shape on held-out test data across eCLIP, iCLIP, and miCLIP experiments. Furthermore, it allows for direct inference of predictions across variable-length sequences and whole transcripts, while outperforming state-of-the-art RBP-binding classification models with respect to the identification of crosslink sites derived from PureCLIP [38], a single-nucleotide peak caller. By performing model interpretation with Integrated Gradients, we demonstrate the capability of RBPNet to accurately identify the sequence patterns driving RBP-RNA interactions and enable binding motif discovery. Lastly, we show the high potential of RBPNet in scoring the impact of single-nucleotide genetic variants on RBP binding, and thereby enable prioritization of functional variants.

## Results

### RBPNet predicts crosslink count distribution from RNA sequence

RBPNet is a deep convolutional sequence-to-signal neural network for modeling protein-RNA interaction profiles, which takes as input a RNA sequence and outputs a probability vector of the same length, describing a discrete distribution of counts within that sequence. In the context of eCLIP, iCLIP, or other individual nucleotide CLIP technologies, RBPNet predicts the distribution of cDNA truncation events, as a result of protein-RNA crosslinking and hereafter also referred to as "crosslink counts," along an input RNA sequence. In this study, RBPNet was trained and evaluated on a large cohort of eCLIP, iCLIP, and miCLIP datasets. An outline of the study is shown in Fig. 1a, which gives a schematic overview of training data generation, model training and interrogation for investigating sequence determinants of RBP binding, including binding motif extraction.

The RBPNet model architecture consists of two major parts——the model body, comprised of the input layer followed by several convolutional blocks with residual connections, and the model head, which performs the final mapping of the input sequence representation, derived from the body model output, to a probability vector (Fig. 1b). Importantly, while RBPNet is trained on fixed-length inputs, its purely convolutional architecture enables prediction on RNA sequences of arbitrary length. During training, the predicted probability vector is used to parameterize a multinomial distribution of crosslink counts and, given the position-wise observed counts in the input sequence

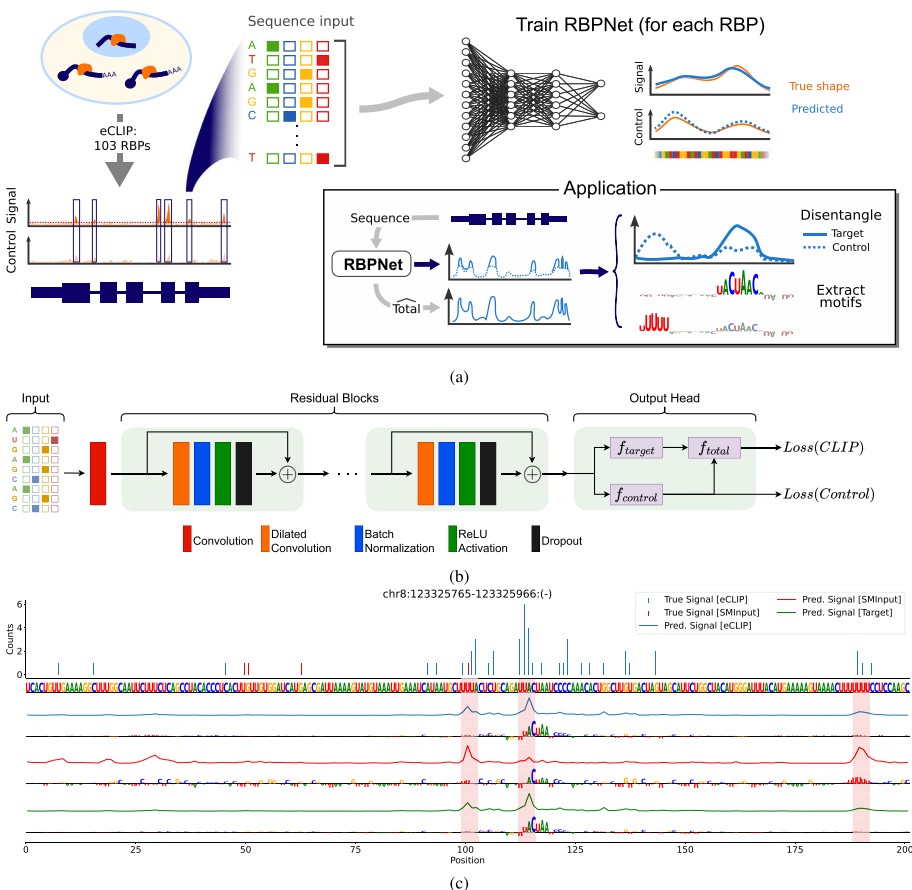

**Fig. 1** RBPNet overview. **A** Schematic outline of data preparation, RBPNet training, and downstream applications. **B** RBPNet model architecture. The one-hot encoded RNA input sequence is first passed through a 1D convolutional layer, followed by several residual blocks, each consisting of a dilated convolution, batch normalization, ReLU, and dropout, respectively. Probability vectors of the *target* and *control* tracks are predicted from the output of the last residual block via a transposed convolutional layer while the *total* track is given by an additive mixture of *target* and *control* tracks. Given the predictions, a loss is computed by taking the sum of the negative log-likelihoods of the observed total and control counts. **C** Example prediction of an RBPNet model trained on eCLIP data of QKI showing observed counts (top) and predicted count distributions for the total (blue), control (red) and target (green) tracks. Integrated gradients feature attribution maps with respect to each predict track are shown below

interval, a negative log-likelihood loss is computed. In other words, the model is penalized in cases where it is unlikely that the observed crosslink counts were drawn from the distribution predicted by the model. RBPNet thus learns the shape of the crosslink count signal, which is subject to the RNA sequence under the assumption that RNA sequence composition is a driver of recognition (and subsequent binding) by RBPs. Similar to other CLIP-based protocols, eCLIP is known to be subject to experimental biases, for instance as a result of enhanced photoreactivity of single-stranded uridine (U) nucleotides during UV-radiation or contamination of eCLIP libraries with other RBPs [20]. Importantly, these biases are sequence-dependent and directly affect the distribution of cDNA truncation counts, hindering the identification of genuine sequence determinants of RBP binding. For that reason, eCLIP experiments are paired with a size-match input (SMInput) control experiment which omits the protein-specific immunoprecipitation

(IP) step, therefore capturing background crosslinking signal from other RBPs or technical biases. To prevent pattern learning of unspecific background and bias signal, RBPNet models the crosslink signal of the control experiment alongside the eCLIP signal. Specifically, RBPNet attempts to explain the observed eCLIP signal as a mixture of two signal components——the *control* component, which is explicitly learned from the control experiment, and an unobserved *target* component, which represents the protein-specific signal (the "RBPNet bias correction" section). The *total* signal, that is, the count distribution of the eCLIP experiment, is then given by a weighted sum of the two components. This is illustrated schematically in the RBPNet output head in Fig. 1b as well as in Additional file 1: Fig. 1a, which outlines the network forward pass in the output head. The mixture of the two signals (target and control) is parameterized by a coefficient $\pi$, which is predicted from the RNA sequence and ranges between 0 and 1. Importantly, we control for technical bias in CLIP assays which we modeled with an additive mixture. This is in contrast to BPNet, a model for chromatin-immunoprecipitation assays which is dominated by DNA accessibility and sequence preference biases that were modeled as multiplicative noise [2].

### RBPNet disentangles bias and protein-specific signal

The formulation of the *total* eCLIP signal as an additive mixture allows for disentanglement of the *target* from the *control* signal, where the predicted *target* signal represents in theory the bias-free, protein-specific crosslinking signal. This is exemplified in Fig. 1c, which shows the observed eCLIP and SMInput read start counts (top), as well as the *total*, *target*, and *control* signal predictions (bottom) using an RBPNet model trained on eCLIP data from the QKI RBP. The RBPNet *total track* (blue) captures well the experimental eCLIP read-start count profile, where the highest enrichment of eCLIP counts can be observed at position 113 of the sequence, immediately upstream of the known QKI binding motif (U)ACUUA [7]. Disentangling the predicted signal with RBPNet shows that the enrichment at this position is mostly attributed to the target, i.e., the true QKI-specific binding signal (green). On the other hand, the experimental eCLIP profile harbors two regions with lower enrichment of read start count around relative positions 102 and 189. Disentangling of the RBPNet total signal reveals that the count enrichment in these regions likely originated from experimental bias, as these regions coincide with elevated signal predictions of the control track (red). Further investigation of RBPNet predictions via Integrated Gradients (IG) [58] feature importance scores with respect to each signal track revealed that the known QKI binding motif (U)ACUUA [7] is correctly recovered in the IG map of the target track, corroborating the evidence that the predicted target signal shape corresponds to the bona fide QKI binding signal. In contrast, a degenerate QKI motif is observed in the IG maps of the control track, while the recovery of U-rich sequence motifs at the modes of the predicted control track distribution further strengthen the observation that those regions correspond to experimental bias. Note that while the SMInput experiment omits the immunoprecipitation, subsequent size-selection for the target protein still leads to a modest enrichment of protein-specific crosslink signal [65]. Therefore, the QKI-specific signal is partially captured in the bias track. While the predicted *total* track is a weighted average of *target* and *control* signal (with a mixing coefficient of 0.92, such that the target is dominating over the control

track), genuine QKI binding signal is correctly recovered by the predicted *target* track. Likewise, signal enrichment mainly representing experimental bias are recovered by the *control* track, while being present in very low proportions in the *target* signal.

### RBPNet predicts eCLIP signal shape at replicate-level accuracy

We next performed evaluation on 103 RBPNet models trained on data from ENCODE [62] eCLIP experiments. To this end, for each eCLIP experiment, candidate training pairs of 300 nt long sequences and crosslink count footprints were first generated without the use of a peak caller via lenient signal-thresholding (the "Methods" section) and subsequently split chromosome-wise into train, validation and test sets (the "Methods" section). Overall, we obtained an average of 302,752 candidate sites across eCLIP datasets, with a minimum of 7937 sites for LARP7 and a maximum number of 1, 105, 807 sites for HNRNPC. Subsequently, RBPNet models were trained separately for each RBP for at most 50 epochs, while the validation loss was observed for the purpose of early-stopping (the "Methods" section). Example predictions on hold-out samples with highest total observed counts for TIA1, QKI, and U2AF2 are depicted in Fig. 2a. Predictions show a high Pearson correlation coefficient (PCC) with the observed eCLIP counts (0.763, 0.830, and 0.872 for TIA1, QKI, and U2AF2, respectively), demonstrating that RBPNet can recapitulate the eCLIP signal shape at high accuracy. Next, we quantitatively assessed correlation of predicted and observed signal distribution. Figure 2b shows the average PCC of RBPNet (total track) predictions with observed eCLIP counts on hold-out samples versus the average PCC of counts between the two eCLIP replicates on the same sequence intervals. RBPNet achieves an average PCC between 0.200 (SSB) and 0.587 (HNRNPC), with an average PCC of 0.328. Strikingly, RBPNet prediction appear to outperform replicates, which have an average PCC of 0.149 across all RBPs. The reason for this are twofold. First, correlation between RBPNet predictions and observed counts are computed with respect to the merged count of both replicates, which reduces sampling effects and thus may increase PCC. Second, RBPNet predicts the count-generating distribution in the given interval, conditioned on the RNA sequence. In contrast, the observations in each replicate represent samples from the true (but unknown) count-generating distribution. As the estimated signal distribution by RBPNet approaches true distribution, the expected PCC between RBPNet predictions and a sample exceeds the expected PCC between the two samples (Additional file 1: Supplementary Text).

### RBPNet enables whole-transcript inference and recovers single-nucleotide resolution binding sites

We next leveraged RBPNet's ability of performing prediction of RNA sequences of arbitrary length, despite being trained on fixed-length inputs. We explored first whether RBPNet can infer signal on entire transcripts by first selecting genes from GENECODE (Release 40) [14] from hold-out chromosomes and subsequently performing RBPNet predictions using models for all ENCODE RBPs. Figure 2f and g show RBPNet predictions (total track) on ENSG00000173207.12 and ENSG00000137955.15 for QKI and HNRNPC, respectively. Indeed, RBPNet predictions show a high correlation with the observed eCLIP counts (0.645 and 0.816, respectively), demonstrating that RBPNet

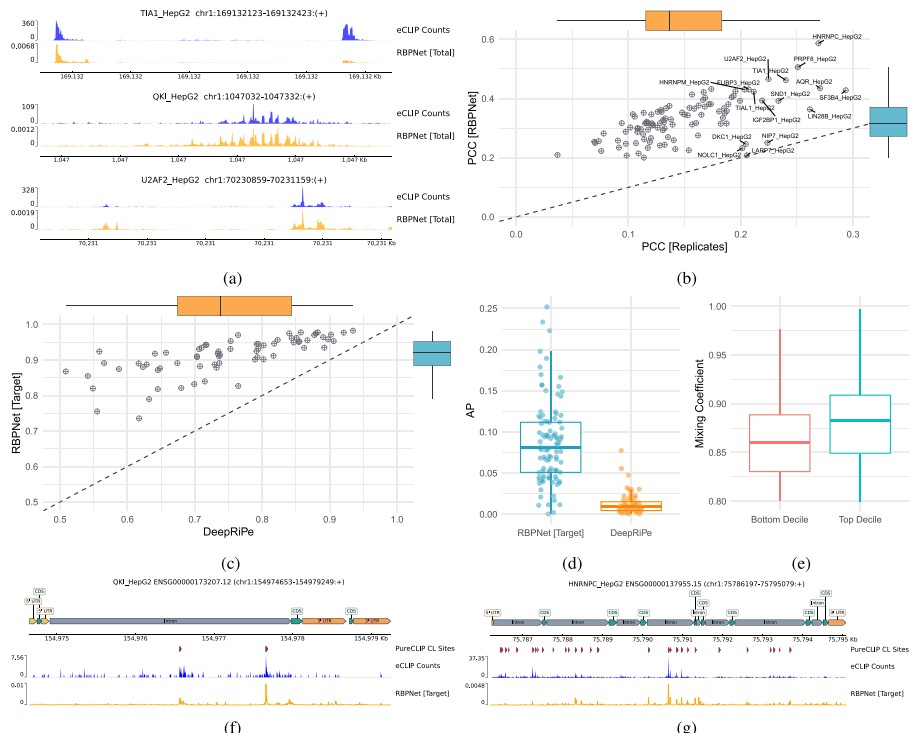

**Fig. 2** RBPNet prediction performance on ENCODE eCLIP datasets. **A** RBPNet predictions (total track) on the highest-count hold-out samples for TIA1, QKI, and U2AF2. **B** Pearson correlation coefficient (PCC) of RBPNet predictions (total track) with observed eCLIP crosslink counts on hold-out samples vs. PCC of observed counts between the two eCLIP replicates. **C**, **D** Mean auROC and AP of RBPNet (target track) and DeepRiPe predictions with respect to crosslink and non-crosslink positions called by PureCLIP across transcripts from hold-out chromosomes, respectively. As pre-trained DeepRiPe models are available only for 70 (out of 103) ENCODE HepG2 RBPs, performance comparison is shown only for those RBPs. **E** Distribution of RBPNet mixing coefficients of the top and bottom decile ENCODE narrow peaks, sorted by eCLIP signal fold-change over the SMInput. High-affinity ENCODE narrow peaks show on average higher mixing coefficients compared to low-affinity peaks. **F**, **G** Example RBPNet (target track) whole-transcript predictions on ENSG00000173207.12 and ENSG00000137955.15, together with observed eCLIP counts and called PureCLIP peaks, for QKI and HNRNPC, respectively

models trained on rather short, fixed-size inputs generalize well to the task of whole-transcript prediction.

Given that RBPNet predictions show high correlation with observed eCLIP counts, we next assessed whether high-scoring RBPNet predictions coincide with peaks called by PureCLIP [38], a single-nucleotide peak caller that identifies significant crosslink sites from eCLIP and SMInput cDNA truncation counts using a Hidden Markov Model. To this end, we performed whole-genome peak calling with PureCLIP on ENCODE eCLIP datasets (the "Comparison with PureCLIP crosslink sites" section), identifying on average 46,459 crosslink (CL) sites per RBP, with a minimum and maximum number of CL sites of 1083 and 585,772 for NIP7 and HNRNPC, respectively. For each RBP, hold-out chromosome genes were intersected with PureCLIP CL sites and transcripts harboring at least 10 CL sites were selected for downstream evaluation. Subsequently, whole-transcript signal shape prediction was performed via RBPNet on selected transcripts. As PureCLIP peaks were called using both eCLIP and SMInput background signal information (the "Methods" section), predictions of the *target* track were used for RBPNet

evaluation. For each transcript, auROC and average precision (AP) performance metrics were computed by treating positions at PureCLIP CL sites as positives and all other positions as negatives (Additional file 1: Table S1). Note that for this evaluation, Pure-CLIP peaks are considered to be the ground truth. Since RBPNet predicts the distribution of crosslink sites along genes, the probability at each position correlates with the genes length as well as the abundance of RBP binding on the transcript, rendering position-wise RBPNet predictions uncomparable across transcripts. Therefore, auROC and AP metrics are computed within transcripts and later averaged across transcripts in order to report the final, RBP-specific auROC and AP scores. In order to assess how the RBPNet behaves with respect to classification-based models, we next compared RBP-Net predictions to DeepRiPe, a state-of-the-art classifier for prediction of protein-RNA interaction, by using a sliding-window approach to obtain pseudo single-nucleotide resolution scores (the "Methods" section). Since pre-trained DeepRiPe models for HepG2 ENCODE datasets were obtained directly from Ghanbari et al. [18], sequences in our hold-out set may have been present during training of DeepRiPe.

Figure 2c and d show the average auROC and AP scores, respectively. RBPNet outperforms DeepRiPe on all RBPs in terms of auROC performance, with an average auROC of 0.89 and a minimum and maximum auROC of 0.58 and 0.98 for SSB and HNRNPK, respectively. In contrast, DeepRiPe achieves a significantly lower average auROC of 0.74. Interestingly, RBP-wise auROC scores of RBPNet and DeepRiPe are strongly correlated (PCC = 0.72). This may suggest that some ENCODE eCLIP libraries are of lower quality or that RNA binding of some RBPs is more difficult to predict from sequence, possibly due to a lack of sequence binding motifs. Notably, RBPNet shows a lower variance of auROC performance across RBPs. Given that classification-based models such as DeepRiPe rely heavily on proper categorization of RNA sequences into binding and non-binding for training, this may indicate a higher robustness of RBPNet due to its training approach, which does not rely on upstream peak calling or labeling of RNA sequences. Both RBPNet and DeepRiPe AP values across RBPs range between 0.00032 and 0.2518 and .00038 and 0.0774, respectively, with RBPNet significantly outperforming DeepRiPe (average AP of 0.086 vs. 0.012, respectively). The generally low AP values of both methods is due to AP being sensitive to class imbalance, with the random AP baseline being equivalent to the fraction of positive samples in the dataset. Here, hold-out transcripts have an average PureCLIP CL site fraction of 0.0014 across RBPs, given that transcripts are expected to harbor orders of magnitude more non-CL than CL sites. Thus, while having low AP, RBPNet and DeepRiPe outperform the random baseline by a large margin. Overall, these results show that RBPNet is a powerful discriminator of PureCLIP CL and non-CL positions across the transcriptome, outperforming state-of-the-art classifiers.

### RBPNet mixing coefficient captures relative eCLIP and SMInput signal abundance

RBPNet models the *total* eCLIP signal as a mixture of protein-specific *target* and *control* signal, weighted by a mixing coefficient which determines the fraction of *target* signal in the *total* signal (the "Methods" section). Additional file 1: Fig. 3a shows the distribution of average mixing coefficients on hold-out samples across eCLIP experiments, which range between 0.038 and 0.943 for proteins EXOSC5 and SFPQ, respectively, with a

mean of means of 0.665. Furthermore, we observed that the distributions of mixing coefficients are generally uni-modal and narrow (Additional file 1: Fig. 4), suggesting that for the majority of experiments, the mixing coefficient is centered around a value that may be interpreted as an experiment signal-to-noise ratio. Indeed, the average mixing coefficient captures the similarity of eCLIP and SMInput experiment counts across hold-out samples ($PCC = -0.395$; Additional file 1: Fig. 3b), demonstrating that experiments with a lower predicted mixing coefficient (and thus a higher weight on the control track) show a higher similarity between the eCLIP counts and counts in the paired control.

To further evaluate whether the mixing coefficient captures the fraction of target and control signals, we inspected mixing coefficients of high-and low-affinity ENCODE narrow peaks [61]. To this end, for each RBP, we obtained ENCODE narrow peaks and ordered them decreasingly with respect to their log2 fold-change (logFC) of eCLIP signal over the SMInput. Peaks in the top and bottom deciles were then selected and extended up-and down-stream from their 5′ end, which generally corresponds to the crosslink site, to yield 300 nt windows. Subsequently, RBPNet predictions were performed for each window and mixing coefficients were obtained. Figure 2e shows the distribution of mixing coefficients on top and bottom decile ENCODE narrow peaks. Indeed, top decile peaks receive on average significantly higher mixing coefficients compared to bottom decile peaks ($p < 2.2 \times 10^{-16}$), suggesting that the RBPNet mixing coefficient can separate high-affinity from low-affinity sites, where the latter contains a higher proportion of background signal.

### RBPNet generalizes to iCLIP and miCLIP experiments

RBPNet may be trained on any genomic sequence with a corresponding nucleotide-wise count signal. To demonstrate that RBPNet generalizes to other CLIP-based protocols, we trained RBPNet on data derived from miCLIP and iCLIP experiments.

miCLIP enables in vivo identification of m6a RNA modifications at single-nucleotide resolution by incubating and subsequently crosslinking extracted RNA with a m6a-specific antibody [43]. After digestion of the covalently bound antibody, reverse transcription often truncates at a remaining polypeptide at the crosslink site, with pileups of truncation events yielding an m6a count signal across the transcriptome. miCLIP data from HEK293 and mESC cells was gathered from Kortel et al. [37] and processed similar to eCLIP data (the "Methods" section), before selecting candidate sites for training and evaluation (the "Methods" section). Subsequently, RBPNet models were trained on both cell lines using a similar architecture and hyperparameters as in eCLIP RBPNet models (the "Methods" section). While the mESC miCLIP experiment was paired with a knockout (KO) control experiment, no control experiment was available for HEK293. We therefore trained RBPNet on HEK293 miCLIP data without *target* and *control* modeling, that is, RBPNet was tasked to predict a single track describing the distribution of *total* miCLIP counts in the HEK293 cell line. RBPNet (total track) showed high signal correlation on miCLIP counts in both cell lines (Fig. 3a), reaching PCCs of 0.51 and 0.48 for HEK293 and mESC, respectively. To evaluate the ability of RBPNet to recover miCLIP single-nucleotide peaks, we again performed peak calling with PureCLIP (the "Methods" section), yielding a total of 2,011,704, and 278,311 for HEK293 and mESC miCLIP, respectively. We hypothesize

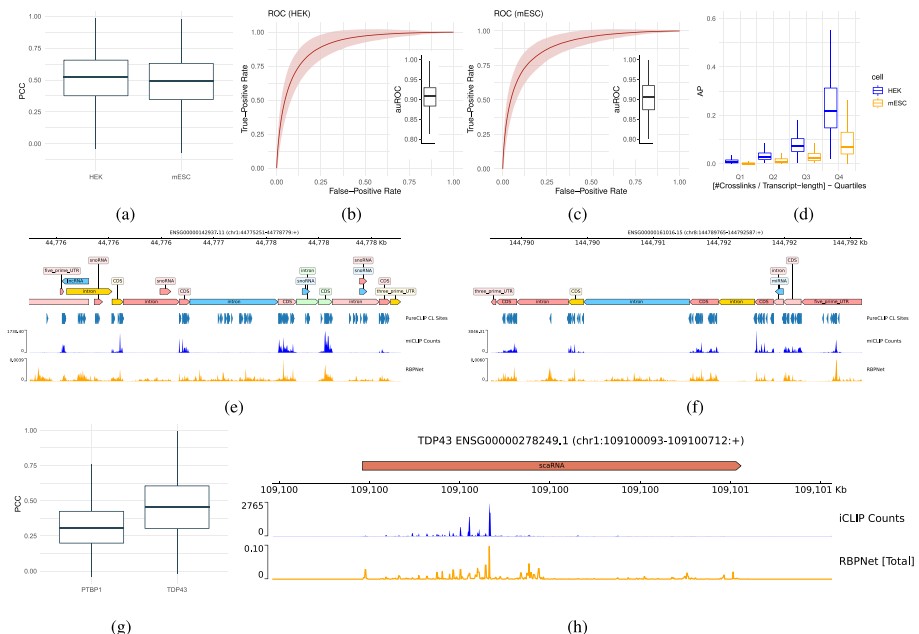

**Fig. 3** Evaluation of RBPNet on iCLIP and miCLIP data. As in contrast to mESC, no control signal is used for m6a peak calling on HEK data, we consequently used RBPNet target and total track predictions for auROC and AP computation for mESC and HEK, respectively. **A** RBPNet PCC performance on hold-out samples of miCLIP experiments in HEK293 and mESC cell lines. **B**, **C** ROC performance of RBPNet whole-transcript predictions with respect to crosslink and non-crosslink positions called by PureCLIP for HEK293 and mESC cell lines, respectively. **D** AP performance on HEK293 and mESC in across transcripts for different PureCLIP crosslink site frequency quartiles. **E**, **F** Example RBPNet-HEK293 miCLIP predictions on ENSG00000142937.11 and ENSG00000161016.15. Notably, both predicted signal shape and observed miCLIP signal occurs predominantly in CDS and UTR regions. **G** Test set PCC performance on iCLIP experiments for PTBP1 and TDP43. **H** Predicted RBPNet signal shape for SCARNA2 (ENSG00000278249.1)

that the significantly higher number of PureCLIP CL sites in HEK293 may be due to a lack of control signal, which in context of mESC may lead to a large number of candidate CL sites being discarded. While the KO control is used for PureCLIP background normalization for the mESC cell line, no control is used for peak calling on the HEK293 cell line, as this was not available. Due to the lack of controls, PureCLIP yields a significantly higher number of CL sites in HEK293 compared to mESC, where a high portion of them may correspond to false positives or noise. Similar to PureCLIP analysis in eCLIP, we then performed whole-transcript inference of miCLIP signal shape on genes of the hold-out set. Figure 3b and c show the ROC curves for HEK293 and mESC, respectively. RBPNet performs well in terms of auROC on both cell lines, with an average auROC of 0.89 for HEK293 and 0.88 for the mESC cell line. In contrast, RBPNet achieves an AP scores of 0.1 and 0.043 for HEK293 and mESC, respectively, with AP performance naturally increasing together with the fraction of positions harboring PureCLIP CL sites. This is illustrated in Fig. 3d, which shows the distribution of AP scores across transcripts grouped into quartiles based on their CL fraction. We consequently observe a lower AP score for mESC compared to the HEK293 cell line as a result of the lower number of PureCLIP CL sites in mESC. Figure 3e shows example RBPNet predictions (total track) on two human genes (ENSG00000142937.11 and ENSG00000161016.15, respectively) from the hold-out

chromosomes using the HEK293 RBPNet model. Interestingly, RBPNet predictions (as well as observed signal) predominantly occur in coding regions (CDS) and 5'/3' untranslated regions (UTRs), while being only slightly present in introns, which is in line with previous study reporting that $m^6A$ methylation mainly those genomic regions [33, 45].

We next evaluated the performance of RBPNet on individual-nucleotide CLIP (iCLIP) data [36]. Compared to eCLIP, it makes use of extra circularization and linearization steps, which allow all cross-linked cDNA fragments to be amplified and sequenced, and a quality control step to assess specificity of pulled-down protein-RNA complexes. To this end, we gathered iCLIP data from Hallegger et al. [23] and Haberman et al. [20] for TDP43 and PTBP1 proteins, respectively, which was processed as described in the "Data and preprocessing" section. As no paired control experiment was available for either RBP, we again trained RBPNet by omitting modeling of *target* and *control* signal and instead tasked RBPNet with predicting the total iCLIP count distribution directly. Figure 3g shows the distribution of PCCs on test set samples for the PTBP1 and TDP43 models. RBPNet reaches an average PCC of 0.46 for TDP43 and 0.320 for PTBP1. Notably, RBPNet reaches a comparable PCC of 0.366 on the PTBP1 eCLIP dataset, which may suggest that the fraction of PTBP1 signal (and thus RNA-binding) explained by RNA sequence is comparable in the eCLIP and iCLIP datasets. This demonstrates the capability of RBPNet to achieve high predictive performance of the RBP-RNA interaction signal shape, independently of the protocol used to generate the data the model is trained on. In order to qualitatively assess the ability of the RBPNet-TDP-43 model to predict signal shapes on full length transcript when trained on iCLIP data, we manually investigated the prediction profile on the hold-out transcript with the highest absolute counts. Figure 3h shows TDP-43 RBPNet predictions and observed iCLIP counts for ENSG00000278249.1 (SCARNA2), a scaRNA associated with DNA repair pathway regulation that has been previously described to be interacting with TDP43 [5, 30]. Indeed, the profile predicted by RBPNet strikingly reflects the observed signal.

**Sequence attribution maps capture RBP-binding motifs**

Recognition of target RNAs by RBPs is in part driven by local sequence features, also known as binding motifs. The identification of RBP-binding motifs is crucial for understanding RBP target recognition and the regulatory grammar present in RNA sequences. While deep learning models were long regarded as black boxes, recent feature attribution methods, such as Integrated Gradients (IG) [58], allow for the identification of input features that contributed significantly to the observed model prediction. In the context of RBPNet, these methods "attribute" a given prediction to nucleotides in the input RNA sequence by assigning a score to each position. Nucleotides that were primarily responsible for the observed crosslink count distribution, such as those residing in binding motifs, receive a higher score compared to nucleotides that did not contribute towards protein-RNA crosslinking. IG attribution maps may be computed with respect to any of the three output track, i.e., control, target and total. Attributions of the target track are

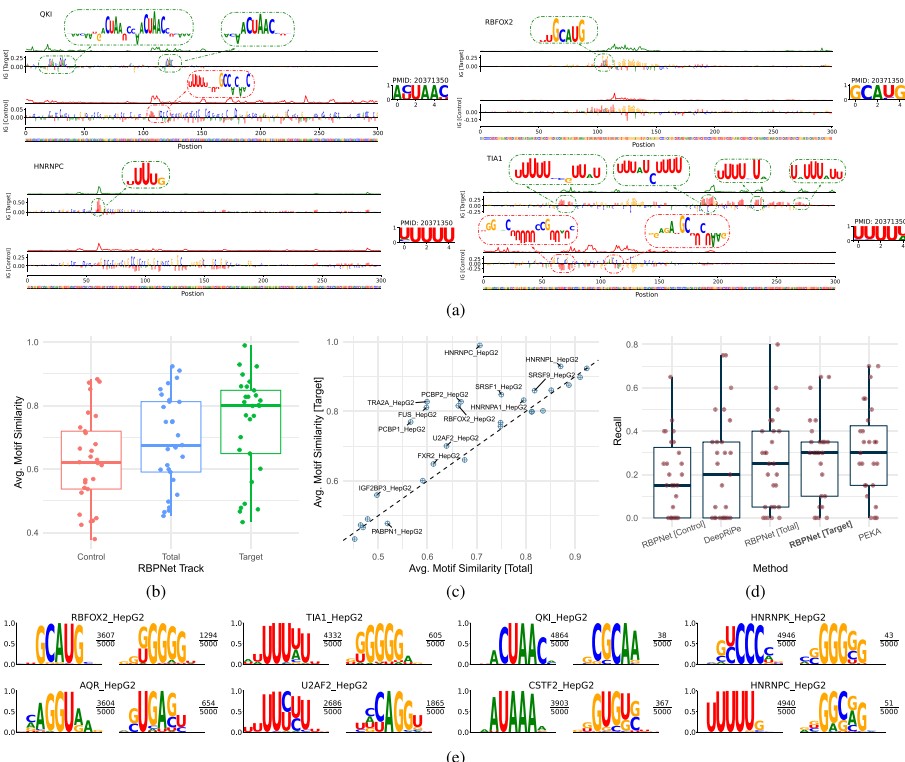

**Fig. 4** RBPNet feature attribution maps and binding motif discovery. **A** Example integrated gradient attribution maps with respect to the *target* track for RBFOX2, HNRNPK, TIA1, and QKI with corresponding motifs taken from the RBPmap database. **B** Distributions of similarity scores between 5-mers extracted from RBPNet attribution maps and PWMs of motifs reported in RBPmap for *control*, *target*, and *total*. **C** Average RBPmap PWM similarity across RBPs for 5-mers extracted from *target* and *total* track attributions. While similarities of the *target* track on par or higher compared to the *total* track for the vast majority of RBPs, the improvement is more pronounced for some RBPs. **D** Recall of the top 20 in vitro 5-mers recovered by the top 20 5-mers extracted from attribution maps of RBPNet *control*, *target*, and *total* tracks as well as DeepRiPe and the PEKA motif finder. RBPNet *target* track and PEKA show comparable performance, outperforming DeepRiPe and the RBPNet *total* track. **E** Consensus motifs constructed from extracted 5-mers of the RBPNet *target* track. Consensus motifs with highest (primary) and second highest (secondary) k-mer support are shown. The corresponding k-mer support is shown as a fraction of the total number of extracted 5-mers next to the consensus motif logos

expected to highlight nucleotides that contributed significantly towards protein-specific crosslinking, while attribution of the control track are expected to explain the unspecific background and bias signals[1].

Figure 4a shows examples of IG attribution maps, computed with respect to *control* and *target* tracks for test-set samples using eCLIP RBPNet models for RBFOX2, HNRNPK, TIA1, and QKI, alongside PWMs of consensus motifs reported in literature, obtained from the RBPmap database [50]. The corresponding predicted *control* and *target* signals are shown above the attribution maps (in red and green, respectively). Target track IG attribution maps show the presence of highly predictive sub-sequences that correspond to known binding motifs for each of the shown RBPs. For instance, IGs of

---

[1] Note that not all technical CLIP bias is manifested in the RNA sequence and thus learnable by the RBPNet model. Therefore, only the sequence-component of the bias will be captured in control track IG maps.

HNRNPC show the characteristic U-motif, while three distinct canonical ACUAAC motifs can be seen in the IGs of QKI. In contrast, control track attribution maps do not show the presence of clear binding motifs, with a general attribution score enrichment at G and C nucleotides. We next performed a global quantitative assessment of how well RBPNet attribution maps can recover known binding motifs across RBPs in the ENCODE eCLIP database. To this end, we selected the top 5000 ENCODE narrow peaks for all ENCODE eCLIP experiments with trained RBPNet models and computed IG attribution maps with respect to *target*, *control*, and *total* tracks for each RBP. For each attribution map, we extracted 5-mers with highest sum IG scores (the "Methods" section). Extracted 5mers were then compared with position-weight matrices (PWMs) of literature motifs obtained from the RBPmap [50] database by computing the similarity between each 5mer and its corresponding RBPmap motif (the "Methods" section). In total, 29 out of the 103 ENCODE RBPs with RBPNet models were represented with at least one PWM in RBPmap, which were then selected for downstream analysis. Figure 4b shows the distribution of average similarity scores of extracted 5mers to RBPmap PWMs across RBPs for each of the three RBPNet prediction tracks. 5-mers extracted from attribution maps of the *target* track show the highest RBPmap-similarity, consistent with the fact that this track represents the de-biased, protein-specific signal predictions, followed by the *total* and *control* tracks, respectively. Notably, on average 5-mers extracted with respect to the control track have the lowest similarity to known RBP PWMs, highlighting the ability of RBPNet to extract different sequence representation for true signal and bias, respectively. To investigate the degree of improvement that signal de-biasing in the *target* track offers over the *total* track for individual RBPs, Fig. 4c shows the average *target* and *total* track RBPmap-similarity for each RBP. Interestingly, we find that while for some RBPs, the *target* track leads to no or only modest improvements of RBPmap-similarity compared to the *total* track, other RBPs, such as HNRNPC, PCBP2, RBFOX2, and TRA2A, appear to benefit strongly from the signal de-biasing of the *target* track. This may reflect variable levels of experimental bias across eCLIP datasets.

### RBPNet IG attribution maps recover in vitro binding motifs

In vitro experiments on protein-RNA interactions, such as RNA-Bind-N-Seq and RNAcompete, offer an orthogonal view to validate motifs identified from in vivo data, as they do not harbor crosslinking specific biases or contamination of experiments with other RBPs and therefore measure intrinsic affinity of RBPs to RNA in an unperturbed environment [41, 52]. To examine whether modeling of the control signal as an auxiliary task can increase the specificity of predicted CLIP signal, we cross compared 5-mers previously obtained from RBPNet IG attribution maps of *target*, *control*, and *total* tracks with 5-mers enriched in corresponding RNA-Bind-N-Seq (RBNS) or RNAcompete in vitro datasets. In total, we evaluated 27 eCLIP datasets in the HepG2 cell line, for which either RBNS (16 RBPs) or RNAcompete (11 RBPs) data was also available.

To measure the agreement between in vitro 5-mers and RBPNet 5-mers obtained in the previous section, we first computed the sum of $IG_{sum}$ scores for each unique 5-mer as a measure of importance with respect to CLIP signal shape prediction. We then calculated the RBPNet recall for each track by taking the fraction of the top 20 in vitro 5-mers

that were recovered in the top 20 5-mers by RBPNet on eCLIP datasets (the "Methods" section). We found that across evaluated RBPs, the RBPNet *target* track recovered a significantly higher proportion of relevant in vitro 5-mers from eCLIP than the *total* track, suggesting that RBPNet can successfully increase the specificity of eCLIP signal (Fig. 4d). As expected, the *control* track recovered the least in vitro k-mers; however, for some RBPs, even the *control* track alone could retrieve high ranking in vitro motifs. This effect could be explained by a partial enrichment of RBP-specific signal in the control experiment, as suggested by a previous study, which evaluated the effect of using eCLIP narrow peaks in contrast to SMInput-agnostic peak-calling on discovery of relevant binding motifs from eCLIP data [39]. Next, we set out to assess whether the governing sequence features learned by RBPNet could be reliably used for motif discovery. To this end, we compared the RBPNet recall to positionally-enriched k-mer analysis (PEKA), a state-of-the art tool for discovery of enriched k-mers from individual CLIP datasets [39] and DeepRiPe. In contrast to RBPNet, PEKA models background from intrinsic crosslinking signal and does not use SMInput controls; therefore, it provides valuable orthogonal view in how specificity of the motifs can be impacted by different background modeling approaches. Surprisingly, we found that the RBPNet *target* track recovered a similar proportion of relevant in vitro k-mers compared to PEKA, despite the fact that RBPNet was not originally designed for objective of motif discovery (Fig. 4d). Lastly, we investigated whether a direct modeling of eCLIP signal, compared to binary labels (bound/unbound) assigned to entire RNA sequences, offers an advantage in the context of motif discovery. To this end, we compared RBPNet recall performance to DeepRiPe by extracting 5-mers from DeepRiPe attribution maps (the "Methods" section). Indeed, both RBPNet *target* and *total* track outperform DeepRiPe in terms of in vitro k-mer recovery (mean recall of 0.294, 0.254, and 0.222, respectively), suggesting that direct modeling of raw eCLIP signal improves binding motif discovery irrespectively of bias modeling. Furthermore, these results indicate that enriched in vitro k-mers are strong predictors of true CLIP signal and that RBPNet learns to associate the presence of these k-mers with high count signal. Lower agreement of the RBPNet *total* track (compared to *target* track) with in vitro k-mers suggests that a fraction of the *total* track signal is explained by sequence features not present in in vitro k-mers, which likely correspond various possible technical sources of sequence biases or contaminant signals in CLIP [22, 39].

### Consensus motifs reveal primary and secondary motifs

Having demonstrated that RBPNet successfully identifies predictive k-mers that coincide with in vitro datasets, we next constructed consensus binding motifs to concisely represent the sequence binding preference of each eCLIP RBP. To this end, we select 5-mers previously derived from RBPNet *target* track attribution maps and build consensus motifs by successively aligning 5-mers in the order of highest to lowest IG importance (the "Methods" section). Figure 4e shows consensus motifs together with the fraction of supporting 5mers for selected RBPs. A full list of consensus motifs extracted with RBPNet is displayed in Additional file 1: Fig. 2 and Additional file 2: Table S2. To investigate which motifs derived from RBPNet are known and which ones are novel, we compared RBPNet motifs with motifs reported across existing databases, including ATtRACT [19], RBPDB [8], mCross [11], oRNAment [4], and RBPmap [50]. Additional file 2: Table S2

depicts RBPNet consensus motifs together with their similarity to motifs reported for the same protein in each of the five evaluate databases (the "Methods" section). Primary binding motifs identified with RBPNet agree with previously identified motifs, including RBFOX2, QKI, TIA1, U2AF2, and HNRNPC. We further identify secondary consensus motifs, which may represent alternative binding preferences or co-factor binding (the "Methods" section). In addition, RBPNet suggests novel candidate motifs for several RBPs, including AQR, SAFB, WDR43, and NIP7, for which no motifs exist in any of the evaluated databases. Several RBPs, including RBFOX2, TIA1, and (to a lesser extend) HNRNPK, show a G-rich secondary motif (Fig. 4e), with some showing G-rich primary motifs (BCCIP and CSTF2T; Additional file 2: Table S2). We hypothesize that this may be due to co-factor binding of an RBP with G-motif preference or, as suggested by previous studies, may indicate contamination of eCLIP libraries with another RBP [39, 62]. For this reason, G-rich motifs are to be treated with care, as they may (in general) not represent genuine binding preference of the target RBP. A complete list of consensus motifs derived from RBPNet are shown in Additional file 1: Fig. 2.

### Studying the impact of sequence variants on protein-RNA interaction with RBPNet

RBPs are highly evolutionary conserved and have been associated with an abundance of human diseases, particularly in the context of degenerative disorders [9, 49]. Recently, Gebauer et al. [17] found that over 1000 RBPs are mutated in context of disease, which amounts to > 20% of proteins annotated with disease-associated mutations. Besides altered coding sequences of RBPs, nucleotide polymorphisms in their RNA targets may impact transcript regulation via loss of binding sites. Indeed, Park et al. [49] showed that dysregulation of RNA target sites from RBPs with a diverse set of functions represents is a key driver of psychiatric disorder risk. Therefore, computationally quantifying the impact of variants with respect to protein-RNA interaction at large scale is important for the prioritization of causal variants in context of disease. Classification-based methods have previously been utilized for the scoring of variants by taking the difference in predicted binding probability between the reference and alternative alleles. In contrast, RBPNet predictions are vector-valued, which complicates a direct comparison of both alleles. Here, we investigated the ability of RBPNet to score the effects of sequence variation on protein-RNA interaction by quantifying variant impact as the divergence between the predicted CLIP-seq (target) signal distributions of the reference and alternative alleles. We demonstrate that resulting impact scores can prioritize variants that may be associated with the disruption of protein-RNA interaction.

Figure 5a exemplifies our variant scoring approach on *rs6981405*, an A-to-C transversion within the *DDHD2* gene that disrupts QKI binding and which has been associated with schizophrenic risk [49]. As shown, in silico mutagenesis leads to change in predicted RBPNet target signal around the variant, with a lower signal amplitude in the alternative (ALT) allele compared to the reference (REF), which we quantify as the Kullback-Leibler (KL) divergence between REF and ALT predictions (the "Methods" section), yielding a scalar impact score. Feature attribution maps (Fig. 5a, bottom) of the REF and ALT predictions reveal that *rs6981405* disrupts the well known QKI binding motif ACUAAC [21]. The strongly negative implications of the A-to-C transversion at the last position of the binding motif is correctly detected by RBPNet, as is evident

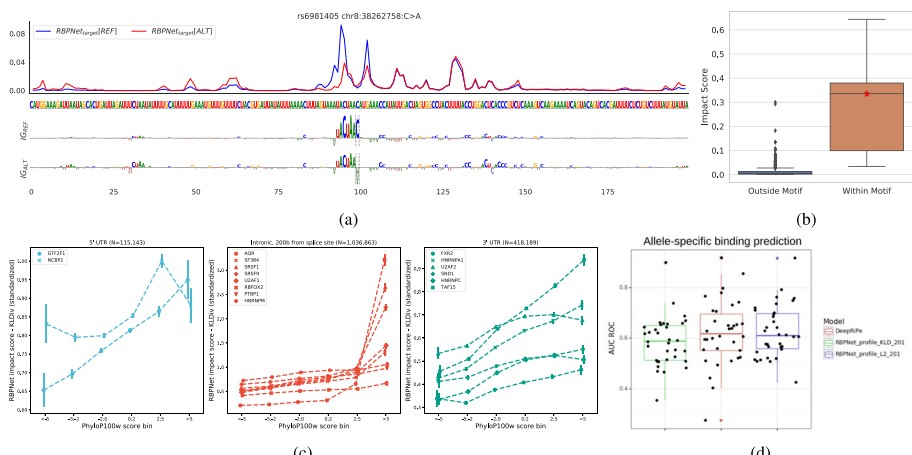

**Fig. 5** RBPNet variant impact prediction. **A** Impact of the *rs6981405* variant on the predicted RBPNet *target* signal. Reference (blue) and alternative (red) signal is shown in a 200-nt window around the variant position. The corresponding feature importance maps for reference and alternative sequence are shown below. A C-to-A transversion at the 5′ end of the QKI binding motif leads to a drastic change of the predicted signal compared to the reference signal, which can be quantified as the KL divergence between the reference and alternative allele predictions. **B** Comparison of impact scores of a systematic in silico mutagenesis of each position towards each of the 3 alternative bases within in a 200-nt sequence window around *rs6981405*. **C** RBPNet impact scores (standardized) on gnomAD variants as a function of sequence conservation, measured via the PhyloP score. Mutations are separated per biotype and their impact is evaluated for RBPs known for biotype-specific activity. Impact scores are higher in regions that are under strong evolutionary constraints. **D** Classification performance of allele-specific binding variants by RBPNet, measured in KLD and L2 norm (the "Methods" section), and DeepRiPe

from the change of high-positive to high-negative attribution at the mutation site. To evaluate whether greater impact is assigned to SNPs within the QKI motif compared to SNPs outside the motif, we performed a systematic in silico perturbation analysis of each nucleotides within a 200-nt window around the *rs6981405* SNP. Figure 5b depicts the distribution of a total of 600 variant impact scores, grouped based on whether they reside within or outside the QKI binding motif. Indeed, the majority of non-motif perturbations lead to small changes in predicted signal profile and thus to small impact scores, while mutations falling within the QKI motifs lead to significantly larger impact scores. Further examples are given in Additional file 1: Fig. 7. This demonstrates that KL divergence is an appropriate metric for scoring the impact of sequence variants based on RBPNet predictions.

### RBPNet variant impact scores are higher in evolutionary constrained regions

To assess whether RBPNet impact scores can identify the disruption of regulatory RNA elements, we investigated the elevation of impact scores in non-coding regions which are under negative selective pressure. To this end, we obtained a set of 1,570,195 SNV from gnomAD [31] which were scored with RBPNet models of a representative set of 15 RBPs, chosen across a broad range of post-transcriptional pathways (the "Methods" section), in order to keep the scoring of variants in a computationally feasibly range. Figure 5c depicts the average impact score (measured in terms of KL divergence) as a function of sequence conservation, measured by PhyloP score across 100 vertebrates, at the variant position. Indeed, impact scores are strongly correlated with sequence

conservation across all investigated RBPs, suggesting that RBPNet identifies the disruption of conserved regulatory RNA elements that engage in protein-RNA interaction. In addition, we investigated the correlation of impact scores with variant allele frequencies (AF) (Additional file 1: Fig. 8). Surprisingly, no clear association between RBPNet KLD impact scores and variant AF could be identified. We speculate that fitness constraints on RNA regulatory motifs may not be prominent enough on population scale, which may render the comparison of impact scores with variant allele frequency inconclusive. In addition, constraints are likely biotype-specific, requiring more sophisticated stratification of non-coding variants which are under negative selection [12].

### Investigating allele-specific binding (ASB) with RBPNet

We next investigated whether RBPNet impact scores can stratify variants that are associated with allele-specific CLIP-seq signal enrichment from a set of background variants. To this end, we obtained and processed variants associated with allele-specific binding (ASB) from Yang et al. [66] (the "Methods" section). Qualitative evaluation of ASB sequences with RBPNet revealed that variants in close proximity may result in false-positive ASB variants, as illustrated in Additional file 1: Fig. 6, which shows 4 ASB-associated SNVs for QKI within a 200-nt interval. RBPNet prediction of each variant allele revealed that only 2 (top left/top right) out of 4 SNVs are associated with a disruption of the QKI-binding motif, which induces substantial changes in the predicted RBPNet signal. The two remaining SNVs (bottom left/bottom right) fall outside a QKI binding motif and thus show only a negligible change of predicted RBPNet signal. KLD impact scores for SNVs impacting QKI motifs are considerably higher (0.28 and 0.12) compared to SNVs outside of motifs (0.05 and 0.02), which is in line with the strong dependence of allele specific effects of SNVs with respect to their distance to known motifs [66] and suggests that variant impact analysis via RBPNet may aid in prioritizing causal SNVs for allele-specific binding.

In order to deplete the set of ASB-associated SNVs for false-positives, SNVs in close proximity ($< 300nt$) were removed from the ASB set. Figure 5d shows the distribution of auROC scores for RBPNet and DeepRiPe [18] with respect to separating ASB and non-ASB SNVs via impact score. While RBPNet can separate ASB from non-ASB SNVs via KL divergence impact scoring ($auROC = 0.59$), DeepRiPe shows a higher performance ($auROC = 0.62$). Interestingly, further evaluations of different RBPNet impact scoring metrics revealed that comparison of reference and alternative allele prediction via the L2-norm of the element-wise difference between the two vectors, yields a performance on-par with DeepRiPe ($auROC = 0.62$). Furthermore, Additional file 1: Fig. 5 shows that performance of RBPNet and DeepRiPe is RBP-specific, suggesting that these methods may complement each other. As several metrics for comparing two vector-valued predictions exist, this suggests that the variant impact scoring of RBPNet may be further improved via systematic metric search and that this metric may be task-specific.

### RBPNet discriminates between functional and non-functional mutations nearby splice junction

Splicing is a complex and tightly regulated post-transcriptional processes in higher eukaryotes that requires concerted binding of multiple RBPs via spliceosomal

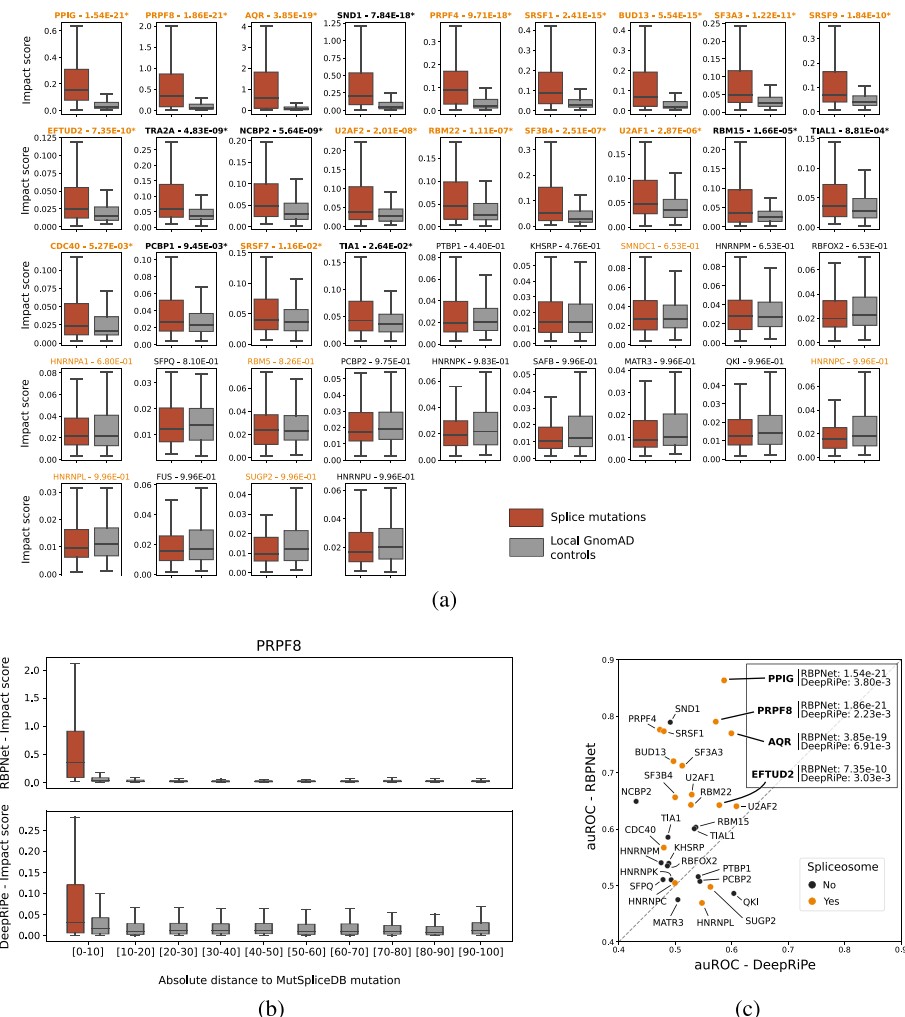

**Fig. 6** Scoring of 232 splicing mutations from MutSpliceDB along with 6087 control mutations from gnomAD taken in their vicinity, using 40 splicing-related RBPNet models. **A** Boxplots of RBPNet impact scores from splicing mutations and local controls per RBP. The 40 RBPs are ordered by their adjusted (Benjamini/Hochberg) *P*-values from Wilcoxon signed rank tests. Title in bold: RBPs with significant *P*-values at $\alpha = 0.05$ (22/40); orange font color indicates the spliceosomal RBPs. **B** Impact score distribution for splice mutations (red boxplot) and gnomAD control mutations (gray boxplots) per absolute, relative-distance bin, with impact scores obtained from RBPNet and DeepRiPe models for the spliceosomal RBP PRPF8 (being the most significant model from DeepRiPe following the Wilcoxon signed rank tests). **C** Scatterplot of area under the receiver operating curve (auROC) calculated from RBPNet models (y) and DeepRiPe models (x). All 30 RBPs annotated as splicing-related in common are depicted, with spliceosomal RBPs in orange. Four RBPs are specifically highlighted for being the only models showing significant *P*-values from the Wilcoxon signed rank tests applied on DeepRiPe models, and their adjusted *P*-values are reported (along the adjusted *P*-values for RBPNet)

complexes, with sequence variants disrupting regulatory binding motifs being associated with severe deleterious effects [15]. To evaluate RBPNet's ability to score the impact of sequence variants on RNA-binding of splicing factors, we obtained a set of 232 splicing-associated mutations from MutSpliceDB [47]. Indeed, we observed greater impact scores for splicing mutations compared to their local controls for 22 out of 40 splicing-related RBPs (Fig. 6a, the "Methods" section). Notably, the set of significant RBPs showed an over-representation of spliceosomal RBPs (15 out of 22), concordant with

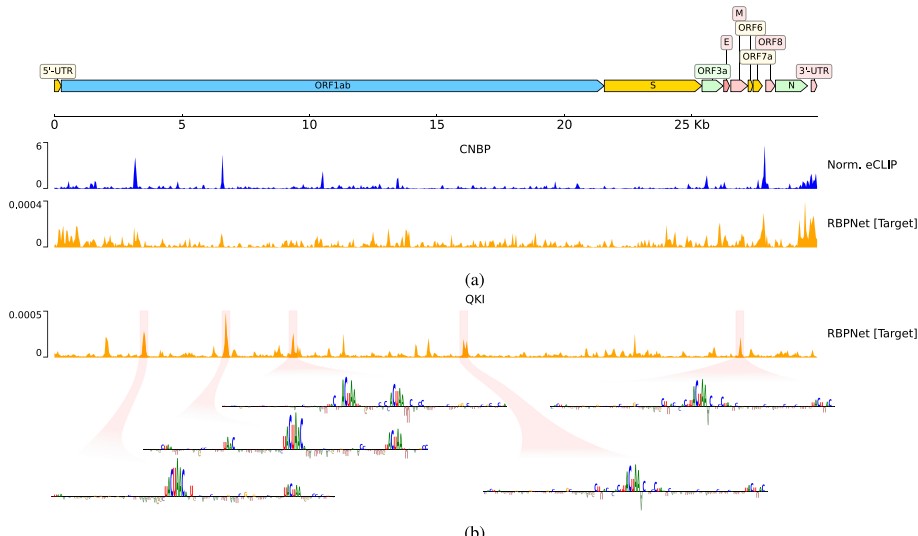

**Fig. 7** Predicting SARS-CoV-2 binding with RBPNet. The top shows target track predictions using a model trained on CNBP eCLIP data, together with the normalized observed signal taken from Schmidt et al. [55]. Shown below are predictions from a RBPNet-QKI model trained on ENCODE eCLIP data, as well as IG attribution maps around the top-5 positions with highest predicted probability

their higher susceptibility to be directly impacted by splicing mutations at splice junctions. Applying the same procedure with DeepRiPe models, only 4 models (out of 30) showed a significant difference in impact scores allowing discrimination between splicing mutations and local negative controls. This is illustrated by the distribution of scores at various relative distance from splicing junctions, as seen for example with the pre-mRNA processing factor 8 (PRPF8, Fig. 6b), a core protein of the spliceosome. While the DeepRiPe model showed the most significant discrimination between functional and non-functional mutations (adjusted $P$-value $= 2.23 \times 10^{-3}$), we can see that the score distribution from non-functional mutations at very short distance ($< 10$ nt) is slightly more elevated than for mutations beyond. On the other hand, the RBPNet models shows a very clear discrimination even at such short distance. We confirmed the improvement of RBPNet over DeepRiPe for scoring variants by comparing the area under the Receiver Operating Curves from each pair of models (Fig. 6c), showing that RBPNet was a better classifier for 26 of the 30 models in common.

### Leveraging RBPNet to infer human RBP binding on viral RNAs

We lastly evaluate RBPNet's ability to infer missing eCLIP signal shape on foreign RNAs. Viruses, including SARS-CoV-2, extensively interact with the host's RBPome [13, 16, 46]. As the large-scale experimental identification of human RBP binding on viral RNA is associated with significant monetary and labor costs, computational imputation of binding sites represents an attractive alternative to identify crucial host factors involved in the virus life cycle. Recently, Labeau et al. [40] experimentally identified binding of QKI to SARS-CoV-2 and reported that QKI-knockout cells are less permissive than wild-type cells.

Here, we utilize RBPNet models trained on eCLIP datasets to predicting binding profiles of human RBPs to SARS-CoV-2 at single nucleotide resolution. Figure 7b shows

RBPNet target track predictions for QKI, showing several high-scoring regions across the SARS-CoV-2 sequence. IG attribution maps at sequence windows around the top 5 highest prediction scores revealed the presence of the QKI-binding motif, suggesting that the RBPNet-QKI model predicts bona fide binding profiles. To further validate that RBPNet predictions on SARS-CoV-2, we obtained eCLIP data for CNBP from Schmidt et al. [55]. Indeed, RBPNet target track predictions correlate significantly with the control-normalized eCLIP signal (Fig. 7a), with an PCC of 0.210. Together, this indicates that RBPNet trained on human eCLIP experiments may be used to extrapolate eCLIP signal shapes to non-human RNA sequences.

### RBPNet webserver

For ease of use, a *BioLib* webserver version of RBPNet is accessible at https://biolib.com/mhorlacher/RBPNet, enabling prediction of signal profiles and sequence attribution maps with pre-trained models used in this study on user-provided RNA sequences.

### Discussion

While high-throughput single-nucleotide CLIP-seq methods offer unprecedented insights into the protein-RNA interaction landscape of RNA-binding proteins, they are limited to transcripts expressed in the experimental cell type at the time of the experiment. Therefore, researchers must rely on computational methods to impute missing binding information on unexpressed transcripts and or foreign RNAs from sequence. In recent years, an abundance of machine learning methods have been developed for the prediction of protein-RNA interaction from RNA sequence, with the most recent iteration of methods relying on deep neural networks to achieve high state-of-the-art performance. However, current classification-based methods assign predicted binding probabilities to the entire input sequence, typically in the order of hundreds of nucleotides, creating ambiguity with respect to the exact location of protein-RNA interaction. Furthermore, the binary labels used in context of classifiers may only contain a fraction of biological information generated by CLIP experiments, which may hinder the learning of complex associations between RNA sequence and RBP-binding. The implicit common goal of protein-RNA prediction methods is a near perfect recapitulation of binding information offered by their in vivo experimental counterparts, which generate nucleotide-wise counts signal across the transcriptome. In this study, we presented a significant milestone towards this goal in the form of a novel deep learning model, RBPNet, which predicts the CLIP-seq signal shape from RNA sequence at single-nucleotide resolution. By training and evaluating RBPNet on eCLIP, miCLIP and iCLIP datasets, we demonstrated that the model is able to predict the experimental signal shape at high accuracy, reaching replicate-level performance and is applicable to data from different CLIP-based protocols. We leveraged RBPNet's ability to processes sequences of arbitrary length at prediction time and impute signal shape on entire gene sequences, some of hundreds of kilo-bases in size, and showed that high-scoring positions predicted by RBPNet coincided with single-nucleotide peaks identified from experimental data via the PureCLIP peak caller. Due to their fundamentally different outputs, establishing unbiased comparisons between RBPNet and classification-based models is challenging, as the later assigns predictions to sequence windows rather than individual positions by design.

To enable comparison with DeepRiPe, a state-of-the-art CNN classifier, we generated pseudo position-wise scores via a sliding window approach and showed that RBPNet significantly outperforms DeepRiPe at recapitulating of PureCLIP crosslink sites. Note that comparison of classifiers (such as DeepRiPe) and methods that directly model the raw experimental signal at nucleotide resolution (such as RBPNet) should be interpreted with care. As RBPNet is the first method to model the nucleotide-wise distribution of CLIP-seq signal as a function of sequence, we compared RBPNet to a current state-of-the-art classification-based method on several tasks (prediction, motif discovery, variant scoring), in order to contextualize RBPNet's performance. However, these models are of very different nature and while both are modeling protein-RNA interaction, RBPNet aims to answer the question of "Assuming there is eCLIP signal, how does it distribute across nucleotides, given the observed sequence?" rather than "Is this sequence expected to generate a signal that passes the peak-calling threshold?". Nevertheless, our results demonstrate that identification of protein-RNA interactions at nucleotide-resolution can not be solved by current classification-based models, stressing the need for new approaches and highlighting the novelty of RBPNet.

Unspecific background signal, as well as experimental biases, is an inherent issue of CLIP-based protocols and thus downstream analysis, as bias towards certain sequence elements confounds the learning of genuine, protein-specific sequence features by modeling approaches. For classification-based models, few strategies have been developed to prevent models from learning bias instead of true signal. For instance, Pysster [6] compiles its negative labeled training set for a given RBP by sampling sequences from binding regions of other RBPs, thus explicitly introducing the same sequence biases of the positive set into the negative set, rendering them non-discriminative for the two classes. Similarly, DeepRiPe performs multi-task learning for several RBPs and only trains on sequences that harbor at least one experimentally identified binding site, such that biases associated with CLIP peaks are present in all input sequences. Yan et al. [67] address the RNase T1 cleavage bias towards guanine in the context of PAR-CLIP and HITS-CLIP by replacing nucleotides in a short window up-and downstream of the input sequence viewpoint with uniformly drawn random nucleotides. In contrast to the above strategies, which exclusively rely on manual manipulation of the training data, RBPNet accounts for experimental bias directly as part of its architecture by modeling the control signal as an auxiliary task. Specifically, RBPNet learns a component which explains the difference in signal shape of the experiment and a paired control, the target track, which is expected to be depleted of experimental bias and instead enriched in protein-specific signal. As the *target* signal is not observed experimentally, we instead quantitatively evaluated the ability of RBPNet tracks to recover known RBP-binding motifs and showed that the *target* track recovers motifs significantly better than the *total* track, with the later modeling the (possibly biased) observed signal. Orthogonal comparison with 5-mers derived from in vitro experiments further showed that the RBPNet *target* track performs comparable to PEKA, a state-of-the-art de novo motif discovery tool. This result is remarkable, as RBPNet is not a motif finder by design and the extracted motifs are derived only in retrospect via model interrogation. Both the RBPNet *target* and *total* tracks outperformed DeepRiPe on the task of in vitro motif recovery, which highlighted the advantage of learning directly on the bulk of raw experimental signal rather than a compressed

representation in the form of binary labels, which may be associated with loss of information. The fact that RBPNet is trained on up to a million (in case of HNRNPC) regions enriched in CLIP-seq signal per RBP may further contribute towards its generalization power. In contrast, classifiers such as DeepRiPe [18] or Pysster [6] are trained on datasets the size of 10–100 thousand samples.

We demonstrated that RBPNet models can be utilized to score the impact of sequence variants on protein-RNA interaction via in silico mutagenesis. For instance, we showed that RBPNet scores single-nucleotide variant within splice junctions significantly higher than randomly selected background variants. In addition, analysis of non-coding transcribed regions revealed that RBPNet variant impact scores are higher in conserved regions, as measured by PhyloP score. This indicates that RBPNet models learned to associated individual nucleotides with the presence or absence of CLIP-seq peaks and that in silico probing of the RBPNet model can improve our mechanistic understanding of protein-RNA interaction.

While RBPNet was shown to predict the count distribution across an input sequence exceptionally well, the lack of a notion of absolute signal hinders a direct comparison of position-wise probabilities between different RNA sequences. This may impose limitation when comparing the binding affinity of two alleles in the context of variant scoring. By design, RBPNet predicts the nucleotide-wise distribution of CLIP-seq signal, rather than the absolute CLIP-seq signal, for a given sequence. Therefore, RBPNet can not natively predict which of two given alleles is expected to have higher CLIP-seq signal. RBPNet may still be utilized for variant scoring by observing that variants which result in gain or loss of CLIP-seq signal will likely alter the distribution of CLIP-seq signal. For instance, a sequence without binding affinity to the target RBP is (in the idealized case) expected to result in the prediction of a uniform CLIP-seq distribution by RBPNet, while a sequence containing a binding motif of the target RBP will result in prediction of a unimodal distribution with high probability mass at the expected crosslinking site. Disruption of the binding motif through a SNV will result in a drastic reallocation of probability mass, which one can quantify. Nevertheless, the lack of absolute binding affinity may hinder RBPNet from accurately ranking the alleles with respect to their binding binding to a target RBP and may explain the diminished performance on tasks such as allele-specific binding scoring, as seen in Fig. 5. Therefore, future work may explore the feasibility of absolute signal prediction, for instance by incorporating transcript abundance coefficients as model covariates. This is a complex task, as different RBPs encounter transcripts at different stages of their life-cycle. Therefore, different estimates of pre-or mature RNA abundance may be required, depending on prior knowledge of the protein at hand. A potential advantage of RBPNet variant scoring is that it can detect instances where a variant alters the *location* of protein-RNA interaction, while not affecting the overall binding affinity of the allele, as such variants induce measurable changes in the predicted CLIP-seq distribution. This indicates that predictions obtained from RBPNet and classifiers (such as DeepRiPe) may complement each other.

Although we demonstrated the power of our approach on several tasks, including CLIP signal shape imputation, identification of bona fide RBP-binding motifs and variant impact scoring, we believe that our results will pave the way for further downstream applications. A promising future application of RBPNet may be in silico peak calling

for the translation of the predicted signal shape into binding sites-a task that is usually performed by peak calling algorithms in the context of experimental CLIP signal. While this task may require prediction of absolute CLIP-seq signal, many peak callers compute position-wise binding site thresholds on a transcript-to-transcript basis (Clipper[2]) or with respect to the local neighborhood (Clippy[3], Paraclu[4]), a strategy that may be adapted for RBPNet.

Avsec et al. [2] demonstrate that the BPNet model may be used to unravel the motif syntax underlying TF cooperativity. In a similar way we envision, as future direction, the application of RBPNet to discover the sequence rules of RBP cooperativity, something that has been only rarely addressed by previous studies. However, CLIP datasets vary greatly in quality, with respect to sequencing depth, number of replicates, replicate consistency, signal-to-noise ratio, and presence or absence of control libraries. While the unique feature of RBPNet to disentangle true signal from noise can in principle enable the accurate identification of composite binding motifs and RBP cooperativity while mitigating the effect of confounding sequence bias, a systematic evaluation of CLIP-seq dataset quality will be necessary to achieve this goal.

## Conclusions

We presented RBPNet, a sequence-to-signal model that predict the distribution of crosslinking events across an input RNA sequence at single-nucleotide resolution. Training and evaluation of RBPNet on 103 eCLIP datasets showed high performance of RBPNet in terms of signal shape correlation, while evaluation on miCLIP and iCLIP datasets demonstrated the models generalization to other CLIP-based protocols. We utilized RBPNet's ability to handle variable-length input sequence to perform inference on whole-transcript and showed that predicted high-probability positions coincide with PureCLIP peaks, outperforming state-of-the-art classifiers. To account for experimental biases, we additionally modeled the signal distribution of paired control experiments and derived a de-biased component, the RBPNet *target* track, which is enriched in protein-specific signal. We showed that feature importance analysis of the de-biased RBPNet *target* yielded informative sub-sequences which recall in vitro motifs at levels comparable to state-of-the-art motif detectors. Finally, we demonstrated RBPNet's ability to score the impact of SNVs on protein-RNA interaction, which may enable prioritization disease-associated variants that disrupt regulatory RNA sequence by causing gain or disruption of RBP-binding sites. RBPNet represents a significant milestone towards full in silico imputation of protein-RNA interaction, while model interpretation suggest that learning on the raw CLIP signal captures more experimental variants, improving our mechanistic understand of protein-RNA interaction.

## Methods

### Data and preprocessing

#### *ENCODE eCLIP*

A total of 103 enhanced CLIP (eCLIP) datasets across 103 RBPs from the HepG2 cell line were obtained from the ENCODE database [62]. Each dataset consists of an eCLIP experiment with two replicates and one size-matched input (SMInput) control experiment, which omits the protein-specific immunoprecipitation step and is thus enriched in unspecific background signal. For each eCLIP and SMInput experiment, aligned R2 reads (i.e., reads whose start positions likely correspond to the position immediately downstream of the RBP cross-linking site) were extracted from the experiment BAM file via SAMtools [42]. Next, reads obtained from the both eCLIP replicates were merged and 5′ read-start coverage for both plus and minus strands was computed via BEDtools [51].

#### *miCLIP*

$m^6A$ individual-nucleotide resolution UV crosslinking and immunoprecipitation (miCLIP) datasets for HEK293T and mESC cells were obtained from Kortel et al. [37], comprising 4 and 2 replicates, respectively. In addition, miCLIP datasets for the mESC cell line are paired with 2 replicates of a METTL3 KO control experiment. For all datasets, bigWig files of crosslink count signals were directly obtained from the Gene Expression Omnibus (GEO) at the accession number GSE163500. Note that bigWig files of reads without C-to-T transitions were selected, as these reads represent read-through events and would result in unspecific truncation count signal. Lastly, replicates of each dataset were merged by summing of the position-wise crosslink counts.

#### *iCLIP*

Individual-nucleotide CLIP (iCLIP) datasets for TDP43 and PTBP1, each with two replicates and without control experiments, were obtained from Hallegger et al. [23] and Haberman et al. [20], respectively. Replicates were downloaded from the Sequence Read Archive (SRA) with accession codes ERS10930255 and ERS10930256 for TDP43 and ERR1588764 and ERR1588765 for PTBP1 and processed as described in [64]. The source code for the processing pipeline is available at https://github.com/ulelab/ncawareclip.

### Selecting candidate sites for training

In order to speed up convergence of RBPNet, it is important to restrict model training to regions with significant crosslink count signal. In the context of BPNet [2], the authors therefore performed peak calling on ChIP-nexus data to select a set of regions highly enriched in count signal. However, recent work by Toneyan et al. [59] suggests that peak callers select sites too conservatively, which may result in under-fitting of sequence-to-signal models.

To train RBPNet on ENCODE eCLIP datasets, we select a large set of candidate sites as follows. Given the set of genes retrieved from the GENCODE (version 40) [14], a sliding window of size 100 is shifted over each gene (*stride* = 1), and the total number of

counts within each window, as well as the highest positional count, is obtained. Next, a *p*-value is computed for each window via a Poisson test by comparing the observed window counts to the expected counts, given the gene-level crosslink counts and the gene-length. At each step, windows with a *p*-value $< 0.01$, a minimum window count of $N = 8$ and a minimum count *height* (i.e., maximum position-wise count within the window) of $H = 2$ are recorded as candidates and the sliding window is shifted forward by 50 nucleotides. This avoids clusters of redundant candidate sites within transcript regions. Finally, selected 100 *nt* windows are extended symmetrically upstrea and downstream to a final length of 300 *nt*. Note that since RNA is a stranded molecule, counts are obtained for each gene in a strand-specific manner.

For miCLIP datasets, similar parameters together with GENCODE vM23 for the mESC cell line where used. For iCLIP datasets, the minimum window threshold was reduced to $N = 4$, due to a lower sequencing depth.

### RBPNet architecture

The body model architecture of RBPNet was inspired by BPNet [2]. RBPNet takes as input a 300-nt RNA sequence, which is one-hot encoded by mapping the bases A, C, G, and U to binary vectors [1, 0, 0, 0], [0, 1, 0, 0], [0, 0, 1, 0], and [0, 0, 0, 1], respectively. The $300 \times 4$ dimensional input is then fed into a 1D convolution layer with 128 filters of size 12, followed by 9 residual blocks. Each residual block consists of (1) a 1D dilated convolution layer with 128 filters of size 6 and exponentially increasing dilation factor, (2) a batch normalization layer, (3) a ReLU activation, and (4) a dropout layer with a dropout rate of 0.25, respectively. The output of the last residual block (hereafter referred to as "bottleneck" layer) then serves as input to one or more output heads, where each output head corresponds to one of the modeling tasks, for instance the prediction of eCLIP signal and (optionally) SMInput shape. This is outlined schematically in Fig. 1b. Each output head consist of a transposed 1D convolutional layer with a single filter of size 20, mapping the bottleneck feature map to a 300-dimensional output vector, which corresponds to the position-wise probabilities of the count distribution within the input window. Notably, as in BPNet [2], same-padding and no pooling is used across all convolution operations in order to conserve the one-to-one correspondence of input sequence positions, feature maps, and outputs.

### RBPNet training

Prior to model training, candidate sites obtained in the "Selecting candidate sites for training" section were split chromosome-wise into validation (chr2, chr9, chr16), hold-out (chr1, chr8, chr15), and train (all other autosomes) sets for both human and mouse cell lines. RBPNet is trained using the Adam optimizer [35] and an initial learning rate (LR) of 0.004. Training is performed for a maximum of 50 epochs with an early-stopping criteria such that training terminates prematurely if the validation loss did not decrease within the last 10 epochs. In addition, a LR schedulers is used such that the LR is halved each time the validation loss did not improve within the last 6 epochs.

## RBPNet loss

The output vector $p_{pred}^h$ of each track $h$ (e.g., eCLIP or SMInput) is used to parameterize a multinomial distribution of read-start counts. For a given training instance, the loss is then computed as the negative log-likelihood of the observed (true) counts $c_{obs}^h$, given the total counts $n_{obs}^h$ in the input region and the probability vector $p_{pred}^h$. That is, the model's loss $L^h$ on a task $h$ is defined as

$$L^h = L(p_{pred}^h, c_{obs}^h, n_{obs}^h) = - log\ p_{mult.}(c_{obs}^h \mid p_{pred}^h, n_{obs}^h) \tag{1}$$

The total loss $L$ is then obtained by taking the sum over all task-specific losses.

## RBPNet bias correction

Experimental bias can lead to unspecific eCLIP signal, severely impacting the downstream binding preference analysis. Therefore, CLIP-seq experiments are usually paired with a control experiment to measure the abundance of background signal at each locus. Assuming that a single read-start count is observed either due to true protein-specific (*target*) signal or experimental bias (*control*), RBPNet models the total CLIP-seq signal as an additive multinomial mixture of the target and bias distributions. That is,

$$p_{total} = \pi \times p_{target} + (1 - \pi) \times p_{control} , \tag{2}$$

where $p_{total}$ is the probability vector of the total (e.g., eCLIP) signal, while $p_{control}$ and $p_{target}$ are the probability vectors of the control and (unobserved) target signal, respectively. Furthermore, $\pi/(1 - \pi)$ is the relative intensity of the target over the bias signal, given by a mixing coefficient $\pi$. Note that $\pi$, $p_{target}$ and $p_{control}$ are learned directly from sequence for each RBP. Note that we explicitly chose to mix target and control signals additively, rather than multiplicatively, as it allows for a clean separation of both tracks. Modeling of biases in the total track multiplicatively requires a non-zero (and possibly large) probability at the respective location in the target track, which may lead to bias artifacts in the target track. To ensure that $p_{control}$ properly approximates the *control* signal distribution over the input sequence, a combined loss on the *total* and *control* tracks is defined as

$$L = L_{CLIP}(p_{total}, c_{CLIP}, n_{CLIP})\ + L_{Ctrl}(p_{control}, c_{Ctrl}, n_{Ctrl}) , \tag{3}$$

such that $p_{control}$ is penalized to match the distribution of counts in the control experiment. Once the RBPNet model is trained we can obtain an approximation of the bias-free signal component as $p_{target}$. A graphical outline of a RBPNet forward pass with bias correction via an additive mixture of *target* and *control* signal is shown in Additional file 1: Fig. 1.

## Estimating the additive mixing coefficient

The contribution of target and bias signal towards the total signal is expected to be dependent on the input sequence. For instance, in the presence of multiple RBP-binding motifs, the majority of counts may be observed due to protein-specific crosslinking,

while under absence of clear binding motifs, crosslinking biases may dominate. RBPNet therefore estimates the multinomial mixing coefficient $\pi$ from the input sequence. Given the feature map of the bottleneck layer (the "RBPNet architecture" section), filter-wise global average pooling is performed along the sequence axis. The resulting 128-dimensional representation of the sequence is then fed into a 1-unit dense layer with linear activation to predict the logit of the mixing coefficient.

### Disentangling target and control signals

Given Equation 2 together with the total number of counts $N$, we can disentangle the eCLIP signal into the expect control and target counts:

$$E_{target} = \pi \times p_{target} \times N \tag{4}$$

$$E_{control} = (1 - \pi) \times p_{control} \times N \tag{5}$$

### Sequence importance scores

To identify RNA sequence features that contributed significantly to the predicted signal distribution, we compute integrated gradients (IG) attribution scores [58] of input sequence with respect to the output probability vector $p$ for each track. This way, we obtain separate attribution maps for predicted total, target and control signals.

By default, the IG attribution method assumes a classification-based setting, where gradients are computed with respect to the output probability of a target class of interest. For instance in the context of classification-based models, attributions may be computed with respect to a single output neuron describing the binding probability of the target RBP to the input RNA. Here, the resulting feature importance values quantify how much each feature contributed towards the target class. For instance, DeepRiPe employs IG to identify nucleotides that were contributed towards predicting an input sequence as "bound" for a target RBP. In contrast to classification-based methods, RBPNet predicts a 1D profile for each RBP and input sequence. Computing IG attribution maps with respect to only a single position in the output track may draw an incomplete picture of nucleotide-wise contributions towards the predicted signal footprint. We therefore introduce a generalization of IG from scalar to 1D profile outputs.

Given an observed input $x$ and a baseline input $x'$, the IG score of an input feature $x_i$ is defined as

$$IG_i(x) := (x_i - x_i') \times \int_{\alpha=0}^{1} \frac{\partial F(x' + \alpha \times (x - x'))}{\partial x_i} \, d\alpha \tag{6}$$

where $F$ is a scalar function. In the simplest case of binary classification, where the deep neural network $f$ has a scalar output, $F = f$. In the case of multi-class classification, $F$ is usually defined as $F(x) = \sum_i p_i y_i$, where $p = f(x)$ is the multi-class probability vector and $y$ the true label vector with $y_i \in {0, 1}$, such that IG scores of $x$ are obtained with respect to its true class. A natural extension of $F$ to count data is given by

$$F(x) = \sum_i p_i c_i \tag{7}$$

where $p$ is the multinomial probability vector and $c$ is the vector of true counts, with the desirable effect that predictions at positions with high counts will dominate the input gradients. The extension of $F$ in (8) has two major drawbacks. First, it requires true counts $c$ for a given sequence to compute attribution scores and second, it might up-weight positions with high counts that are due to experimental bias. We thus reformulate $F$ as

$$F(x) = \sum_i p_i \times stop\_grad(p_i) \tag{8}$$

where $stop\_grad(x)$ stops gradients flow and treats $x$ as a constant.

In other words, instead of computing gradients with respect to the scalar of a single output neuron, we compute gradients of the RNA sequence nucleotides with respect to the sum of the output profile, weighted by a constant version of itself. The weighting ensures that output positions with high probability contribute more towards the nucleotide-wise feature importance scores than low-probability positions.

Note that the proposed generalization of Integrated Gradients to output probability vectors is in analogy to Avsec et al.'s generalization of DeepLIFT [56] scores, described in [2].

By computing gradients with respect to $p_{target}$ (rather than $p_{total}$), we explicitly remove contributions of the sequence towards experimental bias and thus focus solely nucleotides that contribute towards the protein-specific crosslinking signal. In general, attribution scores of the total, target and control tracks may be disentangled via

$$F_{total} = F_{target} + F_{control} = [\pi \times \sum_i c_i p_i^{target}] + [(1 - \pi) \times \sum_i c_i p_i^{control}]. \tag{9}$$

### Performance evaluation

#### *Pearson correlation performance*

Given the set of 300 nt sequences in the hold-out test set, Pearson correlation coefficients (PCC) between RBPNet predictions and the observed crosslink counts, merged between both replicates, were computed. For each eCLIP experiment, the final PCC performance metric is obtained by taking the mean PCC across all test-set sequences.

#### *Comparison with PureCLIP crosslink sites*

PureCLIP is a single-nucleotide peak caller that identifies significant crosslink (CL) sites by fitting a hidden Markov model over the CLIP count signal. To further validate profile predictions made by RBPNet, we investigated whether scores at positions within Pure-CLIP CL sites are significantly higher than scores outside of CL sites. To this end, we performed PureCLIP CL site "peak" calling on all ENCODE eCLIP and miCLIP experiments using default parameters. As suggested by Krakau et al. [38], replicate BAM files

were merged to enable the use of signal information across all replicates during peak calling. For ENCODE eCLIP and the mESC miCLIP experiment, PureCLIP was additionally provided BAM files of the control experiment to refine the set of CL sites based on significant enrichment over the control. This was omitted for the HEK miCLIP experiment, as no paired control experiment was available.

For each dataset, transcripts on the hold-out chromosomes (the "RBPNet training" section) were intersected with PureCLIP CL sites, and only transcripts harboring at least one CL sites were retained, ensuring that the transcript was expressed in the given experiment. Next, whole-transcript predictions were performed with RBPNet, yielding a probability vector of CL enrichment summing up to 1 for each retained transcripts. To measure how well RBPNet predictions discriminate between CL and non-CL sites, the area under the ROC curve (auROC) and the average precision (AP) scores were computed for each transcript. The auROC score may be a more adequate measure of the discriminative power of RBPNet than AP, as the baseline of the AP score is subject to the imbalance of CL and non-CL sites, which are different for each transcript. Furthermore, the auROC is closely related to the Wilcoxon statistic and represents probability of ranking a randomly chosen CL sites above a randomly chosen non-CL site within each transcript, thus directly measuring the discriminative power of the model. Note that within-transcript evaluation is necessary because the position-wise RBPNet scores are subject to transcript length as well as the propensity of RBP binding within the transcript. The finally scalar performance metric for each experiment is then obtained by taking the mean auROC and AP scores across all transcripts.

We additionally evaluated RBPNet against DeepRiPe, a state-of-the-art deep learning model for prediction of protein-RNA interaction, on ENCODE eCLIP datasets. To this end, trained DeepRiPe models for 70 ENCODE HepG2 cell line were obtained from Ghanbari et al. [18]. As DeepRiPe is a classification-based model, the predicted binding probability score is assigned to an entire input regions. To make RBPNet and DeepRiPe comparable on the task of separate PureCLIP CL sites from non-CL sites, we obtained pseudo single-nucleotide resolution scores for DeepRiPe by applying same padding to the transcript sequence, before shifting a sliding window of 150 *nt* (DeepRiPe input size) across the sequence and assigning the prediction score to the center position of the current window.

### Motif discovery and evaluation

#### *RBPmap motif evaluation*
To quantitatively evaluate the ability of RBPNet to recover known RBP-binding motifs in its sequence attribution maps, we compare high-attribution sub-sequences with known binding motifs in the form of position-weight matrices (PWMs) reported in literature. To this end, we first gathered the PWMs of 29 RBPs with both ENCODE eCLIP experiments and reported literature motifs from the RBPmap database [50]. Next, for each eCLIP experiment, the top 5000 ENCODE narrow peaks were selected, and profile predictions were performed on a 300-*nt* window around the 5′ end of the peak, as this position has previously been reported to harbor the CL site [10]. After computing attribution maps with respect to the RBPNet *total*, *target*, and *control* tracks, the 5-mer with highest sum of attribution ($IG_{sum}$) was extracted for each sequence and track. The similarity of

each 5-mer to the reference PWM(s) of the RBP was then computed as the mean of the position-wise Jensen-Shannon divergence (JSD) to base 2, a symmetric version of the KL divergence within the bounds [0, 1], where 1 indicates perfect similarity. To account for cases where 5-mers represent truncated motifs or match the reference PWM at a different position offset (i.e., shifted upstream or downstream), we slide each 5-mer over its reference PWM, with a required overlap fraction of 3 *nt*. At each shift, the JSD is computed and the final similarity of the 5-mer, and the PWM is taken as the maximum similarity over all shift. In cases in which more than one motif PWM is reported for a given RBP in the RBPmap database, similarity computation is performed with respect to all PWMs and the final similarity score is taken as the maximum similarity between the 5-mer and all PWMs of that RBP.

### *In vitro motif evaluation*

In vitro data on protein-RNA interaction was obtained in the form of k-mer *z*-scores for RNA-Bind-n-Seq (RBNS) and RNAcompete experiments from Dominquez et al. [10] and Ray et al. [53], respectively. For RBNS, 5mer enrichment scores (*R scores*) for 78 RBPs were obtained from the ENCODE resource, using accession numbers listed in [10]. For each RBP, the *R scores* for the concentration with the highest enrichment were converted to *z*-scores, by calculating their mean and standard deviation. RNAcompete 7-mer *z*-scores for 80 RBPs were obtained from Ray et al. [53]. In cases where both RNA-Bind-N-Seq and RNAcompete were available for a particular protein, we prioritized RBNS for downstream analysis, as RBNS *z*-scores were readily available for 5-mers, whereas RNAcompete required transformation from 7-mer to 5-mer scores. The conversion of 7-mer scores to 5-mer scores was performed by calculating the mean score across all 7-mers that contain a given 5-mer. 7-mers which contain a particular 5-mer more than once were considered as many times as the number of occurrences of the contained 5-mer. For illustration, when calculating the arithmetic mean of *z*-scores for a 5-mer "UUUUU," the 7-mer "UUUUUUG" would be considered twice ("[UUUUU]UG," "U[UUUUU]G"). In vitro 5-mers of each RBP were then sorted in a descending order based on their enrichment scores and ranked from most (ranked 1st) to least enriched (ranked last).

For evaluation of RBPNet with in vitro 5-mers, 5-mers with highest $IG_{sum}$ were extracted from RBPNet attribution maps of the top 5000 ENCODE narrow peaks for each track, as described in the "RBPmap motif evaluation" section. Furthermore, DeepRiPe ENCODE models were obtained from [18], and unique 5-mer counts were obtained in a similar manner by first computing IG attribution maps on a 150-*nt* input window around ENCODE narrow peaks and subsequently selecting 5-mers of highest $IG_{sum}$ for each narrow peak sequence (the "RBPmap motif evaluation" section). For each unique RBPNet and DeepRiPe 5-mer, a relevance score was then computed by taking the sum of $IG_{sum}$ scores. RBPNet and DeepRiPe 5-mers were then sorted decreasingly with respect to their relevance score. Lastly, we obtained 5-mer enrichment scores calculated with PEKA[5] (v0.1.6), a motif discovery tool, for all ENCODE eCLIP datasets from [39]

---

[5] https://github.com/ulelab/peka

(reference: Additional File 5). We used PEKA-scores that were produced with Clippy peaks [64] to rank the k-mers from most to least enriched. As DeepRiPe models were only available for 70 out of 103 ENCODE HepG2 RBPs, and only 27 of those had orthogonal in vitro data available, the evaluation of recall was therefore restricted to those proteins. Out of 27 eCLIP datasets, 16 were compared to RBNS and 11 were compared to RNAcompete for recall analysis. Finally, an in vitro recall score was computed for each RBP and method by taking the proportion of top 20 5-mers from the corresponding RBNS or RNAcompete dataset that were recovered among the top 20 5-mers in eCLIP, as ranked by the RBPNet tracks, DeepRipe and PEKA.

### Consensus motif construction

Representative consensus motifs for each eCLIP library were constructed as follows. Given the set of k-mers obtained in the "RBPmap motif evaluation" section, k-mers were first sorted by their $IG_{sum}$ score in descending order. Iterating from the top of the list, the first k-mer is used to seed an initial motif alignment, with consecutive k-mers being aligned (without gaps) to the seed k-mer by sliding the given k-mer over the seed alignment and requiring a minimum overlap of 3 *nt*. If no alignment with at least 3 matches is found, the k-mer is considered non-alignable and is instead used to seed a new motif alignment. Consecutive k-mers are aligned to seed alignments in the order of their creation and, if no sufficient alignment is found, are used to seed further motifs alignments on-the-fly. Subsequently, consensus motifs are constructed for each alignment by computing the position-wise nucleotide frequencies within alignments. Consensus motifs can then be prioritized based on the number of supporting k-mers in the underlying alignment. The motif finding procedure was implemented as part of the following Git repository:https://github.com/mhorlacher/metamotif.

### Comparison of RBPNet motifs with existing databases

Motifs were obtained in form of position-weight matrices from the ATtRACT [19], RBPDB [8], mCross [11], and oRNAment [4] databases via RSAT [54] and additionally supplemented with motifs from the RBPmap [50] database. All database motifs were converted to the TRANSFAC format via RSAT convert-matrix. For each eCLIP experiment, the top-2 RBPNet motifs with respect to 5-mer support were selected and subsequently compared to all motifs of the corresponding RBP in each database. As a measure of similarity between two PWMs, the normalized correlation relative to the smallest of the two aligned matrices (NcorS) was computed using the RSAT "compare-matrices" command. As databases commonly report several motifs for a given RBP, the maximum similarity score was taken as the final similarity score between an RBPNet motif and motifs of the corresponding RBP in a database.

### Variant impact scoring

The predicted distribution of counts by RBPNet is solely driven by the input RNA sequence, and thus one expects that single-nucleotide variants (SNVs) that fall within crucial sequence feature, such as binding motifs, will have a profound impact on the predicted signal footprint. Therefore, to approximate the impact of a SNV on RBP binding, we quantify the change of the distribution of counts of the alternative allele compared to

the reference. To this end, we define the change of count distribution with respect to a given SNV as the KL divergence of the prediction on the SNV-associated allele from the prediction on the reference allele. This impact score is thus defined as

$$Impact_{KLD}(SNV) = KLD(p^{REF}, p^{ALT}) . \tag{10}$$

### Scoring of gnomAD variants

GnomAD variants (v2.1, hg38 assembly) with PASS filter status were further filtered in order to keep single-nucleotide variants with available allele frequency, dropping singletons ($AC = 1$). They were then intersected with protein-coding genomic annotations (GENCODE v40) and filtered for mutations within $3'$UTR or $5'$UTR or intronic within 200 nt of a splice site. This resulted in a set of 1570,195 mutations. As a proxy of the negative selection associated to these genomic positions, the mutations were grouped based on their allele frequency, from common mutations ($AF >= 0.05$) to rare mutations ($AF < 0.001$). In addition, they were annotated with the PhyloP 100w score, measuring evolutionary conservation across 100 vertebrates, and grouped so as to separate highly constrained genomic positions (positive PhyloP scores) from neutrally or fast evolving positions (negative PhyloP scores). Finally, a representative set of set of 15 RBPNet models was selected to score the putative impact mutations on RBP binding. RBPs were selected based on genomic region preference and RNA processing functions (obtained from [63]). For instance, selected RBPs are involved in translation regulation (e.g., NCPB2), splicing (e.g., RBFOX2), or post-transcriptional regulation (e.g., SND1). For each RBP, the average and standard error of the impact score was measured for each bin of sequence constraint, evaluated either through the allele frequency or the PhyloP score. Finally, variants were evaluated w.r.t. a positive relationship between the predicted RBPNet impact score and the degree of sequence constraint.

### Scoring of allele-specific binding (ASB) events

Allele-specific binding variants were obtained from Yang et al. [66] and filtered by removing variants with less than 20 reads across both alleles, in order to enrich for a robust set of ASB variants for downstream evaluation. Variants were further filtered by removing all variants with a neighboring variant in a 300-nt. This was done in order to remove potential false-positives (albeit at the expense removing true positives), as variants in close proximity create ambiguity with respect to the causal variant for ASB. RBPs with less than 10 ASB variants were not considered for analysis, resulting in a evaluation set of 44 RBPs. Finally, 10 random positions within the gene of each ASB variant were samples to generate a set of background, non-ASB variants.

### Variant impact scoring of splicing mutations

Forty out of 103 RBPs with trained RBPNet-eCLIP models were manually annotated as related to splicing, following the annotations from Nostrand et al. [63] and the HGNC spliceosomal complex groups from the HGNC database [60]. Of these, 21 were further annotated as directly involved in the spliceosome. Next, a set of 260 experimentally validated

splicing-related mutations was obtained from MutSpliceDB [47]. After excluding mutations with a distance of more than 10 nt from splicing junctions (defined as the first two and last two positions from intronic regions in human coding genes of GENCODE V40 [14]), a set of 232 mutations was retained. For negative controls, we retrieved 6087 mutations from gnomAD v2.1.1 ([31]) located within 100 nt upstream or downstream of the retained splicing mutations. Control mutations which intersected with the set of splicing-associated mutations were filtered out. Subsequently, RBPNet impact scoring (the "Variant impact scoring" section) was performed on all mutations. For each RBP, a one-sided Wilcoxon ranked sum test was performed to evaluate the enrichment of high-impact splicing-associated mutations over control mutations. *P*-values were corrected for multiple testing via Benjamini-Hochberg correction, and significance was tested for $\alpha = 0.05$. The same procedure was applied for 30 of the 70 DeepRiPe models found in common with the 40 RBPNet models, taking the models trained from ENCODE HepG2 using both sequence and genomic annotations. Here, the impact score was calculated as the absolute difference in prediction score for a given RBP between the alternative and the reference allele.

## Supplementary Information

> **Additional file 1.** Includes additional text on the expected variance of replicates, additional figures for the RBPNet architecture, consensus motifs, mixing coefficient analysis and variant scoring analysis.
>
> **Additional file 2.** Includes Fig. S2, a collection of RBPNet-derived consensus motifs and their similarity to motifs of other databases.
>
> **Additional file 3.** Review history.

### Acknowledgements
Not applicable.

### Peer review information

### Review history
The review history is available as Additional file 3.

### Authors' contributions
M.H. and A.M. conceived the project. M.H. collected and processed the datasets and conceptualized and implemented the RBPNet model with help from O.W.; N.W. and L.M. performed the analysis of allele-specific binding events and splice-site mutations, respectively, with help from M.H.; K.K. performed the binding motif recall analysis on in vitro binding data. N.G. helped in processing datasets and training RBPNet models. O.W. and A.M. supervised and guided the project, with inputs from M.S., J.U., and J.G.; M.H. wrote the manuscript with help from K.K., N.W., A.M., and L.M. and inputs from O.W., J.G., and J.U.; all authors reviewed and approved the final manuscript.

### Funding
 This work was supported by the Helmholtz Association under the joint research school "Munich School for Data Science (MUDS)" to M.H., N.W., J.G. and A.M., the Deutsche Forschungsgemeinschaft (SFB/TR501 84 TP C01) to A.M. and L.M. and (SFB/Transregio TRR267) to J.G.; O.W.'s work was funded in part by the Novo Nordisk Foundation through the Center for Basic Machine Learning Research in Life Science (NNF20OC0062606). O.W. further acknowledges support from the Pioneer Centre for AI, DNRF grant number P1; K.K.'s and J.U.'s work was funded by the European Union's Horizon 2020 research and innovation program (835300-RNPdynamics). K.K. and J.U. further acknowledge support from The Francis Crick Institute, which receives its core funding from Cancer Research UK (FC001110), the UK Medical Research Council (FC001110), and the Wellcome Trust (FC001110).

### Availability of data and materials
Code for RBPNet training, evaluation, feature importance analysis, and variant impact scoring is available at GitHub (https://github.com/mhorlacher/rbpnet) [28] and Zenodo (https://doi.org/10.5281/zenodo.8125355) [26]. Code for consensus motif construction is available at https://github.com/mhorlacher/metamotif. Code in both GitHub repositories is available under the MIT license.
All data processed in this study was obtained exclusively from public sources. CLIP-seq data was obtained from ENCODE [62] (eCLIP), Kortel et al. [37] (miCLIP), and Hallegger et al. [23] and Haberman et al. [20] (iCLIP). In vitro data on

protein-RNA interaction was obtained from Dominguez et al. [10] (RNA-Bind-n-Seq) and Ray et al. [53] (RNAcompete). Splicing-related mutations were obtained from MutSpliceDB [47], while further mutations were obtained from gnomAD [31]. Information on allele-specific binding events was taken from Yang et al. [66].

## Declarations

### Ethics approval and consent to participate
Not applicable.

### Consent for publication
Not applicable.

### Competing interests
The authors declare no competing interests.

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

## 
