## [**Additional file 3.** Review history. · Genome Biology]

Review History

First round of review

Reviewer 1

Were you able to assess all statistics in the manuscript, including the appropriateness of statistical tests used? Yes: The statistics is solid and very well presented.

Were you able to directly test the methods? No.

Comments to author:

In this manuscript, Horlacher et al. introduce RBPNet, a sequence-to-signal model that predicts the distribution of cross-linking events across an input RNA sequence at single-nucleotide resolution.

The manuscript is very well written and I think deserves publication in Genome Biology.

All the methods are very well described and the use of controls is particularly key to support the claims of the article. Many different applications are shown: analysis of data without control SMInput, SNVs effects, splicing factors analysis of binding sites and SARS-CoV-2 interactome. The method is available in github. From what I could directly the method is of very broad applicability and easy to use. Comparisons with PureCLIP (from the same lab), DeepRiPe and motif analysis are properly described. This manuscript very cleverly uses neural networks to refine the signal from CLIP experiments.

My comments:

- I wonder why the authors trained the algorithm trained the method on a fixed length of 300 nucleotides. Is there an optimisation behind that?
- I think it could be fair to report also PureCLIP performances in the main images to compare with RBPNet performances.
- It would be very useful to know what is the fraction of newly discovered motifs. I think it is particularly important to have an updated catalogue of motifs for the community.
- I wonder whether some RBPs that are know to form complex together (i.e. protein-protein) show similar binding profiles. It could be very cool to disentangle the signal in eCLIP experiments.

Reviewer 2

Were you able to assess all statistics in the manuscript, including the appropriateness of statistical tests used? Yes.

Were you able to directly test the methods? Yes.

Comments to author:

Summary

Horlacher et al. develop a class of base-resolution RBP binding profile prediction models, called RBPNet, that learns to control for eCLIP technical bias from matched control experiments. The authors train RBPNet on more than one hundred eCLIP experiments in HepG2 from the ENCODE portal, as well as additional iCLIP and miCLIP experiments, and demonstrate robust performance improvements in binding prediction and motif recall relative to state-of-the-art models (e.g. DeepRiPe). Using integrated gradients to produce base-resolution attribution scores of de-biased track predictions followed by 5-mer saliency aggregation, the authors find well-known RBP binding motifs that match motif databases and also identify novel motifs suggesting cofactor RBP binding motifs or alternate binding motifs. Finally, the authors demonstrate the use of their model for predicting and prioritizing genetic variants that interfere with RBP binding motifs, and compare their variant effect scores to DeepRiPe on a set of 232 splice factor-impacting mutations.

The authors clearly demonstrate that their model robustly learns to predict motif binding at base-resolution, out-competing state-of-the-art models based on classification. Since no previous model has attempted to learn RBP binding at this resolution, this tool ought to be of great interest to researchers working with CLIP data. However, the authors' benchmarks on genetic variation are underwhelming and unconvincing. Because such data is compelling for model validation and biological application, we suggest revisions to address this shortcoming.

Major comments

Genetic variation.

The analysis of genetic variation is severely lacking in the manuscript, which makes it difficult to assess how influential the authors' modeling advances will ultimately be for regulatory biology. The authors state that, "we use RBPNet to score the impact of sequence variants on protein-RNA interaction", but then show a single (presumably cherry picked) example of a known disease variant rs6981405, which disrupts QKI binding. The genome is large, and any decent model will lead you to one enticing example. Figure 5B is vapid given the clear recognition of the motif in panel A.

The authors should perform a systematic analysis of genetic variation with their model. I'll make two suggestions, either of which would benefit the manuscript.

1. Predict Gnomad variants and study the relationship between allele frequency and predicted influence on RBP binding. Influential variants should tend to have lesser allele frequencies due to negative selection.

2. Predict fine-mapped eQTLs from a negative set. Many eQTLs will function through transcriptional regulation, so apply some filters to enrich the set for post-transcriptional regulation. See this manuscript for fine-mapped eQTL sets

<https://www.nature.com/articles/s41588-021-00924-w>

Allele-specific binding.

Analogous to the previous argument, the authors' analysis of allele-specific binding is too qualitative and cherry-picked. The authors should perform a systematic benchmark of all of the significant ASB events from Yang et al. Can RBPNet models distinguish between ASB variants and a negative set, chosen to control for some basic properties like genic region, reference/alternative nucleotides, and gene expression? Can the models predict which allele will have greater binding signal for the significant ASB events?

Control track.

RBPNet's strategy to model the control track and combine it with the signal track to match the data appears to be somewhat but not completely effective. E.g. the Figure 1C gradients assign greater scores to the RBP motif than the surrounding sequence. And Figure 4B,D show that the control track does learn RBP motifs for a bunch of the experiments. Thus, we suspect getting this fully right will require more research. It would be helpful to the field if the authors were willing to benchmark alternative approaches, including the following.

- Normalize the CLIP track by the control track in data preprocessing and predict the normalized CLIP track.

- Mix the target and control track predictions multiplicatively instead of additively.

Relatedly, it would be interesting to better understand whether the neural network learns to vary the mixing coefficient as a function of sequence or whether it sticks to the same value for each CLIP experiment and represents a global signal/noise measure.

Minor comments

Figure 5 sequence visualizations are too small to see. Since the far sides of the sequence are less interesting, I suggest truncating the sequences to focus on the center nucleotides.

The RBPmap database may not have it, but the identified motifs for CSTF (described in line 378) are not novel; it is well-known that CSTF binds UGUG-rich motifs. Also, the second identified motif is the core polyadenylation signal AAUAAA. We suggest updating the text to mention this.

We sincerely appreciate the time and effort invested by the reviewers in critically evaluating our manuscript. Their valuable comments and suggestions have significantly contributed to the enhancement of our work. Below, we addressed each comment individually, either by conducting additional analysis, or via elaborative answers. All changes in the manuscripts are highlighted in red, while more comprehensive edits (such as new figures) are additionally referenced in the answers.

Reviewer 1

The manuscript is very well written and I think deserves publication in Genome Biology. All the methods are very well described and the use of controls is particularly key to support the claims of the article. Many different applications are shown: analysis of data without control SMInput, SNVs effects, splicing factors analysis of binding sites and SARS-CoV-2 interactome. The method is available in github. From what I could directly the method is of very broad applicability and easy to use. Comparisons with PureCLIP (from the same lab), DeepRiPe and motif analysis are properly described. This manuscript very cleverly uses neural networks to refine the signal from CLIP experiments.

- *I wonder why the authors trained the algorithm trained the method on a fixed length of 300 nucleotides. Is there an optimisation behind that?*

We did not systematically optimize for the input length. The choice of using a fixed length input rather than variable-length inputs was motivated by the fact that it allows for convenient batching of sequences during model training, as variable-length sequences would need to be padded, arguably leading to suboptimal use of computational resources. Screening literature of classification-based deep learning models for protein-RNA interaction prediction methods showed that the majority of methods use input sizes of around 50-150nt (e.g. iDeepS, PRISMNet, DeepRiPe, GraphProt, RPI-Net). To account for potential long-range interactions, we chose a larger window size of 300nt.

- *I think it could be fair to report also PureCLIP performances in the main images to compare with RBPNet performances.*

We believe there is a misunderstanding here, so we try to clarify. In Figure 2C we show the performance of RBPNet and DeepRiPe in terms of their ability to recover single-nucleotide peaks called by PureCLIP. Here, we treat PureCLIP peaks as ground-truths, while RBPNet and DeepRiPe scores are predicted from RNA sequence alone. As this was not very clear in the previous version of the manuscript, we added a sentence to clarify this (Results section “RBPNet enables whole-transcript inference and recovers single-nucleotide resolution binding sites”).

- *It would be very useful to know what is the fraction of newly discovered motifs. I think it is particularly important to have an updated catalogue of motifs for the community.*

We thank the reviewer for this suggestion. Indeed, a more systematic comparison to existing motif databases would be very interesting for the community. To this end, we gather binding motifs across five additional motif databases, namely ATtract, RBPDB, mCross, RBPmap and oRNament and compare them with motifs identified by RBPNet by computing motif similarity scores using the RSAT package (<http://rsat.sb-roscoff.fr/>). The procedure is outlined in an additional Methods Section titled

“Comparison of RBPNet motifs with existing databases”. We added an additional Supplementary Table 2 (in XLSX format), which indicates which RBPNet motif is covered by which database, as well as the similarity score (computed with RSAT) between RBPNet motifs and motifs of other databases. We discover several novel motifs, i.e. motifs not reported in any of the five databases (highlighted in bold in the XLSX supplementary file). A remark to treat novel G-rich motifs with caution was added, as recent research (Katsantoni et al. 2022, Kuret et al. 2022) showed that those may not represent, in all cases, genuine binding motifs. Finally, we added all motif results, including TSV files of the RBPNet PWMs and PDF of the rendered PWMs, to the RBPNet GitHub repository. A paragraph that briefly discusses our findings was included in the Result text under the Section “Consensus motifs reveal primary and secondary motifs”.

- *I wonder whether some RBPs that are known to form complex together (i.e. protein-protein) show similar binding profiles. It could be very cool to disentangle the signal in eCLIP experiments.*

We agree with the reviewer that the discovery of the *in silico* rules governing Ribonucleoprotein formation is the next task to be tackled by RBPNet. However, as pointed out in the Discussion of the paper, conducting a comprehensive evaluation of binding profile similarity between eCLIP proteins is challenging, partly due to reasons pertaining to the specificity of eCLIP data. For example, analysis of enriched motifs from eCLIP data found that many eCLIP datasets are enriched for similar motifs, especially the G-rich or U-rich motifs, which can be used to calculate the “similarity index”; datasets with high similarity tend to contain features indicating potentially lower technical quality/specificity of data, such as low extent of motif enrichment, lack of canonical RNA-binding domains and lower efficiency of crosslinking (Kuret et al. 2022). Katsantoni et al. also evaluated common motif enrichment patterns in eCLIP, as well as the level of peak overlap between different eCLIP datasets, and similarly found that the datasets with the highest level of peak overlap were enriched predominantly in GC-rich motifs or U-rich motifs (Katsantoni et al. 2022). It is thus possible that eCLIP datasets that share mainly similar profiles reflect contaminant signals of one or few dominant RBPs, rather than biological relationships between the RBPs that were meant to be IPed. One piece of evidence for possible contaminant signal is that the G-rich motifs are enriched even in eCLIP data for RBPs that are not meant to bind to such motifs, such as TIA1, and do not show any enrichment of such motifs in other CLIP data (iCLIP, PAR-CLIP). Moreover, these motifs are particularly prominent in low-quality datasets (Kuret et al. 2022).

RBPNet addresses the issue of common contaminant signal by modeling the SMInput experiments together with eCLIP, to learn the biases present in eCLIP data, which the model represents as a control track. However, the contaminant signal is likely enriched via the immunoprecipitation step, which is lacking in SMInput. Thus, the ability of SMInput to accurately capture specific contaminant experimental signals may be limited if the eCLIP experiments are enriching specific motifs (such as G-rich) to a level above SMInput controls. In such cases, the RBPNet model will attribute such enrichments to the target track as well, and thus contaminant signals could remain strong contributors to commonalities in binding profiles. Indeed, we find that RBPNet target track, which

removes bias present in SMInput controls, still has some enrichment of G-rich secondary motifs for several RBPs that are not known to bind such motifs, namely RBFOX2, TIA1, HNRNPK and HNRNPC (Figure 4e). This is briefly addressed in the manuscript at line 382. Due to this issue, we feel that further experimental work is first needed to study the relevance of overlapping profiles or motif signatures to confidently disentangle technical from biological sources of overlap.

Reviewer 2

Horlacher et al. develop a class of base-resolution RBP binding profile prediction models, called RBPNet, that learns to control for eCLIP technical bias from matched control experiments. The authors train RBPNet on more than one hundred eCLIP experiments in HepG2 from the ENCODE portal, as well as additional iCLIP and miCLIP experiments, and demonstrate robust performance improvements in binding prediction and motif recall relative to state-of-the-art models (e.g. DeepRiPe). Using integrated gradients to produce base-resolution attribution scores of de-biased track predictions followed by 5-mer saliency aggregation, the authors find well-known RBP binding motifs that match motif databases and also identify novel motifs suggesting cofactor RBP binding motifs or alternate binding motifs. Finally, the authors demonstrate the use of their model for predicting and prioritizing genetic variants that interfere with RBP binding motifs, and compare their variant effect scores to DeepRiPe on a set of 232 splice factor-impacting mutations. The authors clearly demonstrate that their model robustly learns to predict motif binding at base-resolution, out-competing state-of-the-art models based on classification. Since no previous model has attempted to learn RBP binding at this resolution, this tool ought to be of great interest to researchers working with CLIP data. However, the authors' benchmarks on genetic variation are underwhelming and unconvincing. Because such data is compelling for model validation and biological application, we suggest revisions to address this shortcoming.

Genetic variation.

The analysis of genetic variation is severely lacking in the manuscript, which makes it difficult to assess how influential the authors' modeling advances will ultimately be for regulatory biology. The authors state that, "we use RBPNet to score the impact of sequence variants on protein-RNA interaction", but then show a single (presumably cherry picked) example of a known disease variant rs6981405, which disrupts QKI binding. The genome is large, and any decent model will lead you to one enticing example. Figure 5B is vapid given the clear recognition of the motif in panel A.

The authors should perform a systematic analysis of genetic variation with their model. I'll make two suggestions, either of which would benefit the manuscript.

We thank the reviewer for their comment and agree that scoring the impact of sequence variants is an important downstream task. Before addressing the reviewer's requests below, we would like to comment on this point. First, we want to point out that our variant impact analysis is not purely qualitative but includes the evaluation of impact scoring across splicing-associated variants (previous Supplementary Figure 3, now moved to Figure 6). Here, we evaluated whether RBPNet variant impact scores can separate 232 splicing-associated variants from 6,087 background variants taken

from gnomAD for 40 splicing-related RBPs. We demonstrated that RBPNet scores splicing-associated variants significantly higher than background variants, outperforming DeepRiPe.

While scoring variants with binary models is usually performed by taking the difference in predicted (scalar) probability of RBP binding by the classifier between the reference and alternative allele, there is, to our knowledge, no established method for performing variant scoring using models with vector-valued outputs, such as RBPNet. Indeed, we believe that RBPNet is the first method of its kind that attempts that, as neither BPNNet or ChromBPNNet propose approaches for variant prediction. As we agree with the reviewers that variant impact scoring is an important application of machine learning models trained on functional genomics and transcriptomics data, we addressed this in the final section of our manuscript and investigated how RBPNet predictions could be utilized for this task. In contrast to scalar binding probabilities, there are many ways to compare vector valued predictions and we believe that a systematic exploration of scoring metrics would be beyond the scope of our, already comprehensive and long, manuscript. Nevertheless, we performed several additional analyses to address the variant impact scoring task, following the reviewer's suggestions. We believe that we now show, quantitatively and at large scale, that RBPNet can score the impact of sequence variation on RBP binding. We do this by (1) demonstrating a positive correlation of RBPNet impact scores with PhyloP conservation scores, (2) better-than-random performance in predicting allele-specific variants and (3) prediction of splicing-associated variants. However, our analysis revealed limitations of RBPNet variant scoring, which we critically discuss in the Discussion section.

We also would like to emphasize that Figure 5A and 5B were intended as a proof-of-concept. The goal of this figure was to demonstrate, using a concrete example, that one can quantify the impact of variants by computing the divergence between the predicted CLIP-seq count distribution of the reference and alternative allele. As this is a novel way of scoring variants, we believe that an instructive example may be required for some readers.

- 1. Predict Gnomad variants and study the relationship between allele frequency and predicted influence on RBP binding. Influential variants should tend to have lesser allele frequencies due to negative selection.*

Following the suggestions of the reviewer, we further evaluate the goodness of RBPNet variant impact predictions on two additional tasks. We evaluated the relationship between RBPNet variant impact scores (measured by KLD) and sequence constraint as measured either by 1. PhyloP evolutionary conservation scores across 100 vertebrates and 2. variant allele frequencies, for 1.5 million single-nucleotide variants obtained from gnomAD. Variant impact was predicted for a set of representative 15 RBPs, which were selected across different mRNA processing functions, such as translation-related or splicing. Please note that this analysis on 1.5 million SNPs is very time consuming and for the time being it could not be performed on all RBPs, due to a temporary (but partially still ongoing) disruption of our IT infrastructure (storage and HPC cluster) due to a cyberattack on our institution in March 2023. Please also note that the 15 RBPs are not cherry-picked but are selected to represent not only different processing functions and preferential transcript binding location (introns, 5' UTRs and 3'UTRs), as highlighted above, but also a broad spectrum of model performances, from high-quality models to less accurate ones, and proteins known to recognize RNAs via defined sequence motifs (such as RBFOX2 or SRSF1) as well as proteins known to bind repetitive stretches of nucleotides or less defined motifs (such as PTBP1 and HNRNP proteins). See Method section "Scoring of gnomAD variants" for further details.

Our findings are presented in a new Result section titled "RBPNet variant impact scores are higher in evolutionary constrained regions". We added a new figure (Figure 5C), which shows that impact scores are strongly correlated with PhyloP sequence conservation across all investigated RBPs, suggesting that RBPNet identifies the disruption of conserved regulatory RNA elements that engage in protein-RNA interaction. Investigation of the relationship between gnomAD allele frequency and variant impact score (Supplementary Figure 8) showed that correlation between AF and impact score are highly protein-specific, with no apparent general association across RBPs. Given the strong correlation between PhyloP and variant impact score, one could expect to see a correlation also there. However, fitness constraints on RNA regulatory motifs may not be prominent enough (or very protein-specific) on population scale and only emerge over longer evolutionary distances. Nevertheless, for completeness and as a reference for future benchmarking applications, we thought it important to also report results of the gnomAD AF analysis in the manuscript.

2. *Predict fine-mapped eQTLs from a negative set. Many eQTLs will function through transcriptional regulation, so apply some filters to enrich the set for post-transcriptional regulation. See this manuscript for fine-mapped eQTL sets <https://www.nature.com/articles/s41588-021-00924-w>*

Generally, as gene expression is dominated by transcriptional regulation, we believe that enriching eQTLs that strictly function through post-transcriptional regulation is difficult. Further, RBPNet generates an impact score for each splicing-associated RBP, while eQTLs are expected to contain variants that disrupt binding of *any* RBP associated with splicing. Therefore, correlating impact scores for a single RBP with eQTL may yield miniscule effect sizes and lead to inconclusive results. Finally, we already assessed the ability of RBPNet to detect splicing variants when applying RBP models of splicing factors (Figure 6), such that this evaluation would be slightly redundant.

Allele-specific binding.

Analogous to the previous argument, the authors' analysis of allele-specific binding is too qualitative and cherry-picked. The authors should perform a systematic benchmark of all of the significant ASB events from Yang et al. Can RBPNet models distinguish between ASB variants and a negative set, chosen to control for some basic properties like genic region, reference/alternative nucleotides, and gene expression? Can the models predict which allele will have greater binding signal for the significant ASB events?

We start by addressing the question raised by the reviewer on whether RBPNet impact scores can predict which allele will have greater binding signal. By design, RBPNet predicts the nucleotide-wise *distribution* of CLIP-seq signal, rather than the absolute CLIP-seq signal, for a given sequence. Therefore, RBPNet can not natively predict which of two given alleles is expected to have higher CLIP-seq signal (i.e. binding affinity). Nevertheless, RBPNet may still be utilized for variant scoring by observing that variants which result in gain or loss of CLIP-seq signal will likely alter the distribution of CLIP-seq signal as well. For instance, a sequence without binding affinity to the target RBP is expected to result in the prediction of a uniform CLIP-seq distribution by RBPNet (in the idealized case). On the other hand, a sequence containing a binding motif of the target RBP will result in prediction of a unimodal distribution with high probability mass at the expected crosslinking site. Disruption of the binding motif through a SNV will result in a drastic reallocation of probability mass. In contrast, a SNV outside a binding motif will result in little to no probability mass reallocation. Measuring the degree of probability mass reallocation, i.e. through KLD between the two alleles, can thus score variant impact without predicting absolute binding affinity to the allele sequences.

As suggested by the reviewer, we performed a systematic evaluation of whether RBPNet can separate ASB from non-ASB binding variants. To this end, we obtained and processed the whole set of ASB variants from Yang et al. Background non-ASB variants were generated by sampling positions within the same gene of each ASB variant (see Methods section "Scoring of allele-specific binding (ASB) events"). The new Figure 5D shows the ASB classification performance of RBPNet and DeepRiPe, measured as auROC per protein. While this demonstrates that RBPNet can separate ASB from non-ASB variants (auROC = 0.59), it is outperformed by DeepRiPe (auROC = 0.62). As KLD was our first metric of choice and we previously did not perform a systematic search for other evaluation metrics, we compared KLD with two other metrics, Jensen-Shannon divergence and L2 norm. Interestingly, use of L2 as a scoring metric put RBPNet on par with DeepRiPe, suggesting that more suitable metrics might exist for scoring variants on the basis of RBPNet predictions and that these may depend on the specific task. Nevertheless, we were surprised that RBPNet variant scoring via KLD falls short of DeepRiPe by an auROC margin of 0.03, despite significantly outperforming DeepRiPe w.r.t. nucleotide-resolution predictions (Figure 2C) and prediction of variant impact of splicing-associated SNVs (Figure 6). We attribute this to RBPNet's lack of absolute binding affinity predictions, which through this analysis, emerged as a potential limitation which we critically discuss in the manuscript and may address in a future study.

Together, we demonstrate that RBPNet can score ASB variants significantly better than random and that further investigation of RBPNet variant scoring schemes, as well as augmentation of the model to incorporate prediction of absolute binding affinity, may further improve performance.

In conclusion, our PhyloP, ASB and splice-variant analyses confidently show that RBPNet can be used to score the impact of functional variants. However, our analysis also highlighted potential limitations of RBPNet variant scoring and we envision improvements that could address these issues in future studies. We expanded the Result and Discussion sections to accommodate these findings. Since RBPNet predicts the distribution of eCLIP counts, conditioned on the RNA sequence, the difference in *absolute* binding affinity between two alleles can not directly be assessed. This is in contrast to previous classification-based methods, which are trained to predict the probability that a given input RNA sequence contains one or more CLIP-seq peaks, which are usually defined by applying a threshold on the abundance of CLIP-seq signal at a given site. We believe that this may explain why RBPNet does not outperform DeepRiPe with regards to ASB variant scoring, despite showing higher performance in terms of single-nucleotide binding prediction across transcripts, motif discovery and splicing-associated variant prediction. It is important to note that a potential advantage of RBPNet variant scoring is that it can detect instances where a variant alters the location of protein-RNA interaction, while not affecting the overall binding affinity of the allele, as such variants induce measurable changes in the predicted CLIP-seq distribution. This shows that predictions obtained from RBPNet and classifiers (such as DeepRiPe) may complement each other.

The additional prediction of absolute eCLIP signal may be explored in future studies, as this would introduce a notion of absolute signal into the RBPNet model. This is a complex task, as the absolute eCLIP signal is subject to transcript abundance. Therefore, any model that aims to predict profiles of absolute eCLIP signal must consider estimates of transcript abundance as co-variants (e.g. derived from RNA-seq data in the same cell lines). Different RBPs encounter transcripts at different stages of their life-cycle, such that different estimates of pre-or mature RNA abundance may be required, depending on prior knowledge of the protein at hand. We added these considerations to the Discussion section.

More generally, we believe that comparison of classifiers (such as DeepRiPe) and methods that directly model the raw experimental signal at nucleotide resolution (such as RBPNet or BPNNet) should be interpreted with care. These models are of very different nature and while both are modeling protein-RNA interaction, RBPNet aims to answer the question of “Assuming there is eCLIP signal, how does it distribute across nucleotides, given the observed sequence?” rather than “Is this sequence expected to generate a signal that passes the peak-calling threshold?”. While we previously emphasized this in the context of prediction performance comparisons of DeepRiPe and RBPNet, we now extended this note of caution to the variant scoring task (see Discussion).

Control track.

RBPNet's strategy to model the control track and combine it with the signal track to match the data appears to be somewhat but not completely effective. E.g. the Figure 1C gradients assign greater scores to the RBP motif than the surrounding sequence. And Figure 4B,D show that the control track does learn RBP motifs for a bunch of the experiments. Thus, we suspect getting this fully right will require more research. It would be helpful to the field if the authors were willing to benchmark alternative approaches, including the following.

By design of the SMInput experiment, it is expected that the control partially captures target-protein-specific signals. During SMInput preparation, the IP step is omitted, however, a size-selection for 75kDa (which roughly corresponds to the expected RNA fragment size) over the target protein size is performed. Thus the resulting control is enriched in fragments that are crosslinked with proteins in the same size-range as the target protein, many of which may very well be the target protein itself. We refer to Figure 3 in *Wheeler, Nostrand and Yeo (2018)*, which nicely demonstrates why the SMInput is partially enriched in target protein signals. On page 4 of the manuscript we had included a footnote that addresses this phenomenon when describing Figure 1C, as indeed, a degenerate QKI motif can be clearly seen in the control track. We removed the footnote and added it (in a modified form) as a paragraph in the main text (lines 165-166).

We want to emphasize here that our goal is to correct our predictions for potential experimental biases and thus, our sole objective is to reduce bias in the predicted *target track*. Isolating biases in a *control track* is a byproduct of our approach and the “purity” of the *bias track* is not indicative of the purity of the *target target*. Figure 4B, C and D show that the *target track* better recovers motifs when compared to the uncorrected *total track*, which demonstrates that the bias correction is successful in that it yields a sizable improvement over the *total track*.

-Normalize the CLIP track by the control track in data preprocessing and predict the normalized CLIP track.

As our method learns the parameters of a multinomial distribution, it needs to operate on count data. Therefore, the current framework is not applicable to real-valued, normalized counts. Note that we explicitly opted for a model that directly operates on the raw count data and does not require any pre-processing. Given that RBPNet is the first method that attempts to model CLIP-seq data directly (and at nucleotide resolution), we believe that evaluating additionally modeling strategies for raw CLIP-seq data (i.e. a benchmark) should be addressed systematically in future studies.

-Mix the target and control track predictions multiplicatively instead of additively.

We initially experimented with both multiplicative and additive mixtures, but observed that in some cases, multiplicative mixing led to artifacts in the *target track* so we did not pursue this direction further. The choice for an additive mixture was additionally motivated by the fact that modeling of biases in the *total track* multiplicatively requires a non-zero (and possibly large) probability at the respective location in the *target track*. For this reason, we opted for additive mixing. While we believe that a comprehensive benchmark of additive vs. multiplicative mixing is out of the scope of

this study, we agree with the reviewer that a systematic benchmark of additive vs. multiplicative mixing across several methods (RBPNet, BPNNet, ChromBPNNet, etc.) would be helpful for the field and could be addressed in future research. We added a paragraph to the main text (Methods, Section “RBPNet Bias Correction”), to better justify the choice of an additive mixing of signals.

Relatedly, it would be interesting to better understand whether the neural network learns to vary the mixing coefficient as a function of sequence or whether it sticks to the same value for each CLIP experiment and represents a global signal/noise measure.

This is an interesting point. To address this, we predicted the mixing coefficients of hold-out samples across all 103 ENCODE HepG2 eCLIP experiments with RBPNet and evaluated its distribution as well as its relationship with model performance. Our findings are summarized in an additional supplementary figure (Supplementary Figure 3) and are discussed in an additional paragraph in the main text (Results, Section “RBPNet mixing coefficient captures relative eCLIP and SMInput signal abundance”). Supplementary Figure 3A shows that the mean mixing coefficient varies strongly across eCLIP experiments, with a maximum and minimum coefficient of 0.943 and 0.038 for protein SFPQ and EXOSC5. Regarding the distribution of sample-wise mixing coefficients, Supplementary Figure 4 depicts the distribution of mixing coefficients for all 103 eCLIP experiments evaluated in this study. We observed that the distributions are generally unimodal and narrow, demonstrating that for the majority of experiments, the mixing coefficient is indeed centered around a value that may be interpreted as an experiment signal-to-noise ratio. Related to this, Supplementary Figure 3B shows that the average mixing coefficient is negatively correlated (PCC = -0.395) with the similarity of the experimental eCLIP and control tracks and positively correlated (PCC = 0.375, Supplementary Figure 3C) with model performance, further indicating that the average mixing coefficient is capturing experiment quality.

While one may argue that based on these findings, the model may be trained with a global experiment-wise mixing coefficient (rather than predicting it from sequence), we do not believe that the added complexity impacts model convergence. In addition, providing the model with the capability to predict the mixing coefficient on a sequence-to-sequence basis may reduce artifacts in samples at the tails of the mixing coefficient distribution. Finally, some datasets may benefit substantially from a sample-wise mixing coefficient. For instance, PRPF8, FAM120A or SND1 show a broad distribution of mixing coefficients, which suggests that the shape of the mixing coefficient distribution varies across experiments.

Figure 5 sequence visualizations are too small to see. Since the far sides of the sequence are less interesting, I suggest truncating the sequences to focus on the center nucleotides.

We thank the reviewer for pointing this out. As Figure 5 was overhauled considerably, previous Figure 5C and 5D were moved to Supplementary Figure 7, while Figure 5E was moved to Supplementary Figure 6 and their sizes were increased. We believe that these Figures are now readable, even without additional truncation.

The RBPmap database may not have it, but the identified motifs for CSTF (described in line 378) are not novel; it is well-known that CSTF binds UGUG-rich motifs. Also, the second identified motif is the core polyadenylation signal AAUAAA. We suggest updating the text to mention this.

We removed the claim that the CSTF motif identified by RBPNet represents a novelty. As Reviewer 1 additionally asked for the fraction of novel motifs among our identified motifs, we additionally performed a systematic and comprehensive comparison of motifs identified by RBPNet with five popular motif databases (ATtRACT, RBPDB, mCross, RBPmap and oRNAMENT). An additional table (Supplementary Table 2) was added, indicating which RBPNet motif is covered by which database, as well as the similarity between RBPNet motifs and motifs of other databases. Further details are outlined in an additional methods section titled “Comparison of RBPNet motifs with existing databases”. Motif PWMs as well as logos of extracted motifs were added to the GitHub repository. This analysis demonstrates that RBPNet motifs agree with motifs of other databases on well studied proteins. At the same time, RBPNet suggests several motifs for RBPs not reported in any of the aforementioned datasets. A paragraph that briefly discusses our findings was included in the Result text under the Section “Consensus motifs reveal primary and secondary motifs”.

Second round of review

Reviewer 2

Overall, the authors have addressed my concerns, and I believe the manuscript is nearly ready for publication. I have one question about the new analysis that I expect could be answered in a minor revision.

Major comment

The PhyloP analysis looks compelling. However, I see that the genomic positions come from several different transcript regions, which I expect have different PhyloP distributions. It's difficult to parse this result when the RBPs tend to prefer to bind different regions. I suggest that the authors perform this analysis separately for each transcript category — 5' UTR, 3' UTR, and intron.

Minor comments

Figure 5c is hard to understand when looking at the figure and caption alone; reading the main text paragraph is required to understand the visualization. I suggest adding more detail about what is plotted in the caption.

I suggest that the authors examine this recent preprint, which adds context to their Gnomad MAF analysis.

Findlay, S. D., Romo, L. & Burge, C. B. Quantifying negative selection in human 3' UTRs uncovers constrained targets of RNA-binding proteins. *bioRxiv* (2022)
doi:10.1101/2022.11.30.518628

Authors Response

Point-by-point responses to the reviewers' comments:

Reviewer 2:

Comment 1: The PhyloP analysis looks compelling. However, I see that the genomic positions come from several different transcript regions, which I expect have different PhyloP distributions. It's difficult to parse this result when the RBPs tend to prefer to bind different regions. I suggest that the authors perform this analysis separately for each transcript category — 5' UTR, 3' UTR, and intron.

Answer: We repeated the analysis for the three biotypes, 5' UTR, 3' UTR, and intron. The figure (Figure 5C) was split into 3 sub-figures, one for each biotype.

Comment 2: Figure 5c is hard to understand when looking at the figure and caption alone; reading the main text paragraph is required to understand the visualization. I suggest adding more detail about what is plotted in the caption.

Answer: We believe that splitting the figure into 3 sub-figures makes the figure easier to parse. We also added a title to Figure 5C.

Comment 3: I suggest that the authors examine this recent preprint, which adds context to their Gnomad MAF analysis.

Findlay, S. D., Romo, L. & Burge, C. B. Quantifying negative selection in human 3' UTRs uncovers constrained targets of RNA-binding proteins. bioRxiv (2022) doi:10.1101/2022.11.30.518628

We thank the reviewer for making us aware of this study. We updated the main text and included a reference to this study in the section "RBPNet variant impact scores are higher in evolutionary constrained regions".